# Structural network alterations in focal and generalized epilepsy assessed in a worldwide ENIGMA study follow axes of epilepsy risk gene expression

Epilepsy is associated with genetic risk factors and cortico-subcortical network alterations, but associations between neurobiological mechanisms and macroscale connectomics remain unclear. This multisite ENIGMA-Epilepsy study examined whole-brain structural covariance networks in patients with epilepsy and related findings to postmortem epilepsy risk gene expression patterns. Brain network analysis included 578 adults with temporal lobe epilepsy (TLE), 288 adults with idiopathic generalized epilepsy (IGE), and 1328 healthy controls from 18 centres worldwide. Graph theoretical analysis of structural covariance networks revealed increased clustering and path length in orbitofrontal and temporal regions in TLE, suggesting a shift towards network regularization. Conversely, people with IGE showed decreased clustering and path length in fronto-temporo-parietal cortices, indicating a random network configuration. Syndrome-specific topological alterations reflected expression patterns of risk genes for hippocampal sclerosis in TLE and for generalized epilepsy in IGE. These imaging-transcriptomic signatures could potentially guide diagnosis or tailor therapeutic approaches to specific epilepsy syndromes.

Epilepsy is characterized by recurrent seizures and affects over 50 million people worldwide[1]. Cumulating evidence in epilepsy research has underscored the importance of interconnected brain networks in understanding the causes and consequences of the disease[2,3]. In the common epilepsies, particularly temporal lobe epilepsy (TLE) and idiopathic generalized epilepsy (IGE), histopathological and neuroimaging studies have demonstrated structural and functional compromise across widespread brain networks[3–7]. Magnetic resonance imaging (MRI) analysis of brain morphology, including cortical thickness measurements and grey matter volumetry, provide in vivo evidence of structural alterations across multiple cortical and subcortical regions in both TLE[8,9] and IGE[10,11]. Beyond small cohort studies, robust patterns of atrophy across widespread brain networks were identified in the common epilepsies through the ENIGMA-Epilepsy (Enhancing NeuroImaging Genetics through Meta-Analysis) consortium, with data aggregated from multiple international sites[12].

Covariation of morphological MRI markers, termed "structural covariance analysis," can extend earlier results on regional mapping of healthy and disease-related structural brain organization by identifying complex network mechanisms. Structural covariance has been associated with several aspects of brain organization and development in both health and disease[13]. In healthy individuals, several studies have shown moderate correspondence with both structural and functional connectivity measures, suggesting partial overlap yet also complementarity of different network mapping techniques[14–16]. By comparing cross-sectional covariance networks to longitudinal changes in neurotypical adolescents, prior work has demonstrated a close association of covariance with maturational networks, suggesting that these networks may reflect coordinated trophic processes across the brain[17–19]. Furthermore, several studies have pointed to a close association between covariance network layout, heritability, and gene expression, suggesting that genetic factors are also likely reflected in covariance network organization[20,21].

Prior work applying graph theoretical analyses to structural covariance networks has also characterized normative network topology[22], revealing the presence of a "small world" organization. This architecture, which incorporates high clustering within segregated communities together with short paths between them, may provide a balance between network specialization and integration[23]. In TLE and IGE, structural covariance studies show syndrome-specific deviations from such a topological arrangement. In TLE, increased path length and clustering has been observed using both whole-cortex analysis[24,25] and in limbic/paralimbic[26] subnetworks. In contrast, diverging topological alterations have been reported in IGE, echoing either global increases in clustering[27,28] or path length[29], global decreases in path length[28], or no changes in network measures[30]. Analysis of structural brain metrics using multi-site data gathered by ENIGMA-Epilepsy provides an opportunity to consolidate network alterations in the common epilepsies in a generalizable manner.

Interactions across multiple spatial scales, ranging from genetic factors to macroscale cortical morphology and structural networks, shape cortical and subcortical organization in both health and disease[31]. When combined, these naturally intertwined dimensions offer new insights into the pathophysiology of system-level disorders such as epilepsy[32]. Neuroimaging studies of large-scale networks can profit from studies on the landscape of genetic risk factors in common epilepsies[33,34]. Recently, the open release of postmortem human transcriptomics datasets, such as the Allen Human Brain Atlas (AHBA), has offered opportunities to explore how gene expression patterns in the brain reflect macroscale neuroimaging findings[35,36]. Integrating imaging and genetics can shed light on the micro- to macroscale mechanisms that contribute to the pathophysiology of the common epilepsies. In parallel, this combination can also be used to understand the ways in which genes may reflect, to some extent, network alterations in epilepsy. How, and whether, structural covariance network properties converge with spatial expression patterns of risk genes for epilepsies, however, remains an unanswered question.

In this ENIGMA-Epilepsy study, we aimed to identify robust structural network disruptions in individuals with TLE and IGE relative to healthy controls, aggregating inter-regional cortical thickness and subcortical volume correlations across 18 international sites. Graph theoretical analysis assessed global and regional topological disruptions in both epilepsy syndromes. Moreover, we leveraged gene expression information from the AHBA to relate macroscale network findings to spatial expression patterns of genetic risk factors in these two major forms of epilepsy. Spatial associations between topological changes and disease-related gene expression maps were evaluated against spatial permutation and "random-gene" null models[37,38]. Reproducibility of our findings was also assessed across sites, variable network construction approaches, and clinical variables (side of seizure onset, disease duration).

## Results

**Data samples**. We studied 866 adults with epilepsy (377 males, mean age $\pm$ SD = 33.82 $\pm$ 9.48 years) and 1,328 healthy controls (588 males, mean age $\pm$ SD = 30.74 $\pm$ 8.30 years) from 18 centres in the international Epilepsy Working Group of ENIGMA[39]. Our analyses focused on two patient subcohorts with site-matched healthy controls: TLE with neuroradiological evidence of hippocampal sclerosis ($n_{HC/TLE}$ = 1083/578, 257 right-sided focus) and IGE ($n_{HC/IGE}$ = 911/288). Subject inclusion criteria and case-control subcohorts are detailed in the Materials and Methods and Table 1. Site-specific demographic and clinical information are provided in Supplementary Table 1. All participants were aged between 18–50 years.

**Inter-regional morphometric correlations**. Cortical thickness (measured across 68 gray matter brain regions[40]) and volumetric data (measured across 12 subcortical gray matter regions and bilateral hippocampi) were obtained from all patients and controls. Cortical and subcortical data were harmonized across scanners and sites using CovBat[41], and statistically corrected for age and sex. Group- and site-specific structural covariance networks were then computed from morphometric (cortical thickness/subcortical volume) correlations (Fig. 1a).

Site-specific correlation matrices in TLE, IGE, and healthy controls exhibited similar patterns, with generally strong correlations between bilaterally homologous regions and strong correlations between regions within the same lobe. Overall mean strength of positive correlations across all density thresholds did not differ between individuals with TLE and controls (all $t < 1.86$, $p_{FDR} < 0.13$), but was increased in individuals with IGE relative to controls (all $t < 2.41$, $p_{FDR} < 0.05$).

**Global network characteristics**. To characterize the topology of structural covariance networks, we computed three fundamental and widely used graph-theoretical parameters[42]: (i) mean clustering coefficient, to quantify local network efficiency, (ii) mean path length, to index global efficiency, and (iii) mean small-world index, to quantify the interaction of both local and global efficiency. Notably, the interplay between clustering coefficient and path length can categorize network topology into regular or random, and consequently assess deviations from an optimal

**Table 1 ENIGMA Epilepsy Working Group demographics.**

| Case-control subcohorts | Age (mean ± SD) | Age at onset (mean ± SD) | Sex (male/female) | Side of focus (L/R) | Duration of illness (mean ± SD) |
|---|---|---|---|---|---|
| TLE ($n = 578$) | 35.89 ± 9.15 | 15.09 ± 11.23[a] | 267/311 | 321/257 | 21.12 ± 13.02[a] |
| HC ($n = 1,083$) | 31.72 ± 8.54 | – | 490/593 | – | – |
| IGE ($n = 288$) | 29.65 ± 8.75 | 14.73 ± 8.55[a] | 110/178 | – | 14.46 ± 10.86[a] |
| HC ($n = 911$) | 29.95 ± 8.18 | – | 385/526 | – | – |

Demographic breakdown of patient-specific subcohorts with site-matched controls, including age (in years), age at onset of epilepsy (in years), sex, side of seizure focus (TLE patients only), and mean duration of illness (in years). Healthy controls from sites that did not have TLE (or IGE) patients were excluded from analyses comparing TLE (or IGE) to controls. [a]Information available in 544/578 TLE patients and 248/288 IGE patients.

small-world architecture. At either extreme, regular, or "lattice-like," networks have high clustering and path length, whereas random networks have low clustering and path length. On the other hand, small-world networks are neither completely random nor regular, but have high clustering and low path length and, thus, reflect a locally and globally efficient organization (Fig. 1b)[43].

Comparing patients with TLE to controls, our multisite analysis revealed modest increases in overall small-worldness ($p_{uncorr} < 0.05$ at network densities ($K$) = 0.05–0.18; see Methods) and clustering coefficient ($p_{uncorr} < 0.05$ at $K = 0.08–0.11$), as well as decreases in mean path length over multiple density thresholds ($p_{FDR} < 0.1$ at $K = 0.05–0.15$, 17–18, 22–23, and 0.30–0.50) in patients. In contrast, IGE patients showed, on average, similar overall small-worldness and clustering coefficient relative to controls, but marginal decreases in overall path length at higher network densities ($p_{uncor} < 0.05$ at $K = 0.31–0.50$ Fig. 1c), possibly targeting weaker interregional correlations.

**Regional network characteristics**. We also quantified clustering coefficient, path length, and small-world network changes at the nodal level. Multivariate comparisons, combining only clustering coefficient and path length in TLE relative to controls, revealed trends for topological alterations in bilateral parahippocampus (left/right $p_{FDR} = 0.052/0.054$), paracentral lobule (left/right $p_{FDR} = 0.052/0.052$), lateral occipital cortex (left/right $p_{FDR} = 0.052/0.052$), putamen (left/right $p_{FDR} = 0.052/0.052$), and caudate (left/right $p_{FDR} = 0.054/0.062$), ipsilateral angular gyrus ($p_{FDR} = 0.052$) and orbitofrontal cortex ($p_{FDR} = 0.052$), as well as contralateral insula ($p_{FDR} = 0.052$), middle ($p_{FDR} = 0.052$), and inferior temporal gyri ($p_{FDR} = 0.053$). Although regional Cohen's $d$ effect sizes estimated across sites revealed an overall increase in small-worldness in TLE, particularly in default-mode regions (Cohen's $d$ mean ± SD = 0.30 ± 0.14; Supplementary Fig. 1a), there were deviations away from this configuration in different subnetworks. Notably, paralimbic and limbic regions such as the bilateral orbitofrontal, temporal, and angular cortices as well as ipsilateral amygdala showed an increase in clustering coefficient and path length (Cohen's $d$ mean ± SD: clustering = 0.20 ± 0.13, path length = 0.18 ± 0.11), suggestive of a more regularized, "lattice-like," subnetwork arrangement (Fig. 2a).

When compared to controls, individuals with IGE showed widespread multivariate topological alterations in left inferior frontal gyrus pars opercularis ($p_{uncorr} = 0.0038$), superior temporal sulcus ($p_{uncorr} = 0.012$), and nucleus accumbens ($p_{uncorr} = 0.0080$), and right calcarine sulcus ($p_{uncorr} = 0.0023$), insula ($p_{uncorr} = 0.0061$), inferior temporal gyrus ($p_{uncorr} = 0.010$), and lateral occipital cortex ($p_{uncorr} = 0.0097$), although these findings did not survive correction for multiple comparisons. Effect sizes for each individual metric revealed decreased clustering coefficient and path length, with predominant changes in bilateral fronto-temporo-parietal cortices, nucleus accumbens, and pallidum (Cohen's $d$ mean ± SD: clustering = –0.15 ± 0.11, path length = –0.22 ± 0.17), suggesting a more randomized network configuration (Fig. 2b). Conspicuous decreases in small-worldness were also observed in IGE, affecting predominantly fronto-parietal (bilateral paracentral lobule, right precentral gyrus) and temporal (left middle temporal gyrus, right inferior temporal gyrus) regions (Cohen's $d$ mean ± SD = –0.27 ± 0.19; Supplementary Fig. 1b).

**Transcriptomic associations**. Having established multivariate topological abnormalities in TLE and IGE, we evaluated whether these network-level findings were associated with the spatial expression patterns of previously established genetic risk factors. To this end, we assessed spatial correlations between epilepsy-related gene expression maps and multivariate topological profiles. Epilepsy risk genes were obtained from a recently published genome wide association study (GWAS) on the International League Against Epilepsy (ILAE) Consortium cohort, which comprised 15,212 epilepsy cases stratified into six epilepsy subtypes (Supplementary Table 2)[33]. Gene expression levels of these epilepsy risk genes were then derived from the Allen Human Brain Atlas and averaged across each epilepsy subtype[35] (see Materials and Methods). Positive correlations between brain maps of multivariate topological findings (from Fig. 2) and epilepsy gene expression levels then indicate a spatial correspondence between changes in network topology and molecular phenotype. Significance of imaging-transcriptomic correlations was established using spin permutation tests (termed $p_{spin}$)[37] that control for spatial autocorrelations from the ENIGMA Toolbox (https://github.com/MICA-MNI/ENIGMA[44];). Additional "random-gene" permutation tests (termed $p_{rand}$) were performed to (i) examine gene specificity[38] and (ii) ensure that imaging-transcriptomic associations were not driven by differences in the number of genes in each syndrome- or disease-specific set (see Materials and Methods and Fig. 3a). We found significant associations between the spatial patterns of multivariate topological alterations in TLE and epilepsy risk gene expression levels of hippocampal sclerosis ($r = 0.33$, $p_{spin} = 0.0028$). On the other hand, multivariate topological changes in IGE were related to the expression levels of generalized epilepsy ($r = 0.31$, $p_{spin} = 0.0032$; Fig. 3b). In both TLE and IGE, imaging-transcriptomic associations remained significant (TLE: $p_{rand} = 0.0030$; IGE: $p_{rand} = 0.018$) when compared against null distributions of effects based on selecting randomized, and equally sized, gene sets from the pool of all available genes from the Allen Human Brain Atlas ($n = 12,668$).

**a** | Construction of group- and site-specific structural covariance networks

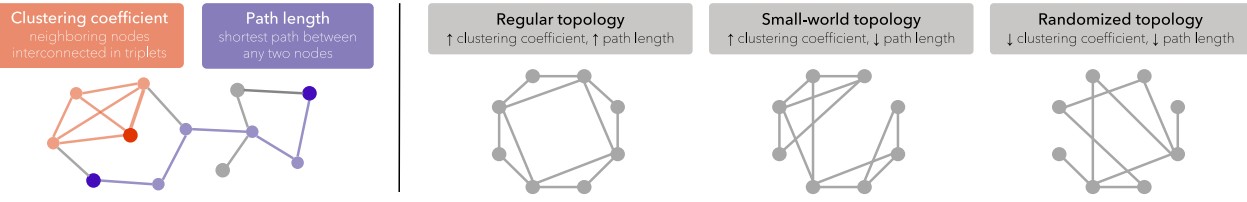

**b** | Graph theoretical parameters and topological properties

**c** | Global network alterations in the common epilepsies

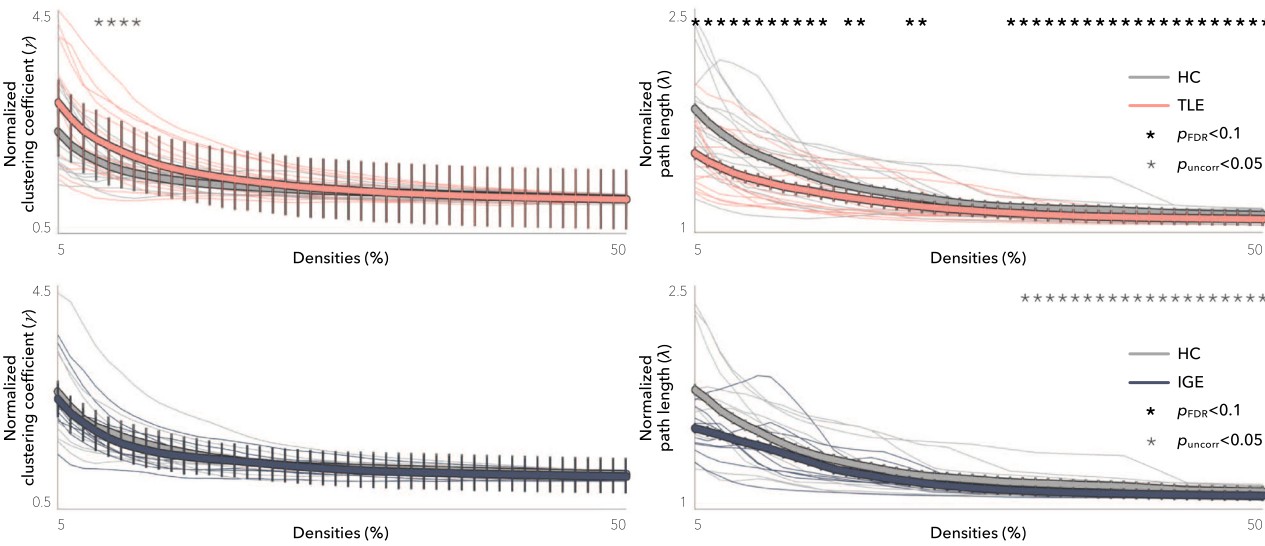

**Fig. 1 Structural covariance networks in the common epilepsies. a** Schematic showing the construction of group- and site-specific structural covariance networks from morphometric correlations. **b** Two graph theoretical parameters characterized network topology: clustering coefficient, which measures connection density among neighboring nodes (*orange*) and path length, which measures the number of shortest steps between any two given nodes (*purple*). The interplay between clustering coefficient and path length can describe three distinct topological organizations: regular networks with high clustering and path length (*left*), small-world networks with high clustering and low path length (*middle*), and random networks with low clustering and path length (*right*). **c** Global differences in clustering coefficient (*left*) and path length (*right*) between TLE and HC (*top*) and between IGE and HC (*bottom*) are plotted as a function of network density. Increased small-worldness (*i.e.*, increased clustering and decreased path length) was observed in individuals with TLE, whereas individuals with IGE showed decreases in clustering and path length, suggesting a more random configuration. Two-tailed student's *t*-tests were performed at each density value, comparing global measures in patients (TLE or IGE) to controls; bold asterisks indicate $p_{FDR} < 0.1$, semi-transparent asterisks indicate $p_{uncorr} < 0.05$. Thin lines represent data from individual sites. Error bars indicate standard error of the mean. HC = healthy control, IGE = idiopathic generalized epilepsy, TLE = temporal lobe epilepsy, $p_{FDR}$ = *p*-value adjusted for false discovery rate, $p_{uncorr}$ = uncorrected *p*-value.

To further assess specificity to hippocampal sclerosis (in TLE) and generalized epilepsy (in IGE) genes, we cross-referenced our network findings with transcriptomic maps derived from (i) genes associated to four additional epilepsy phenotypes, namely: all epilepsy, focal epilepsy, juvenile myoclonic epilepsy, and childhood absence epilepsy[33], (ii) a set of monogenic epilepsy genes from the Epi4K Consortium[45] and the GeneDX comprehensive epilepsy panel (http://www.genedx.com), (iii) genes that are targets of currently used anti-seizure medications[46], and (iv) six

sets of genes associated with common neuropsychiatric conditions and/or comorbidities of epilepsy, including: attention deficit/hyperactivity disorder[47], autism spectrum disorder[48], bipolar disorder[49], major depressive disorder[50], migraine[51], and schizophrenia[52] (Supplementary Table 2). Network alterations in TLE did not correlate to any other epilepsy subtype (range $r = -0.15$–$0.13$, all $p_{spin/rand} > 0.11/0.16$; Fig. 4). In contrast, IGE showed additional significant associations with transcriptomic maps derived from all epilepsy ($r = 0.37$, $p_{spin/rand} = 0.0019/$

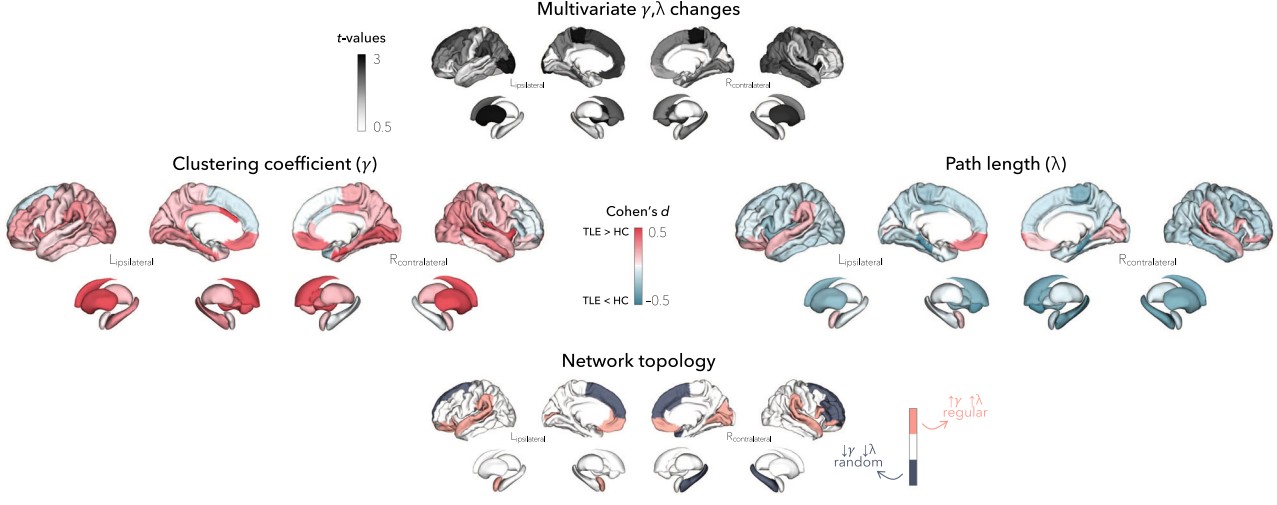

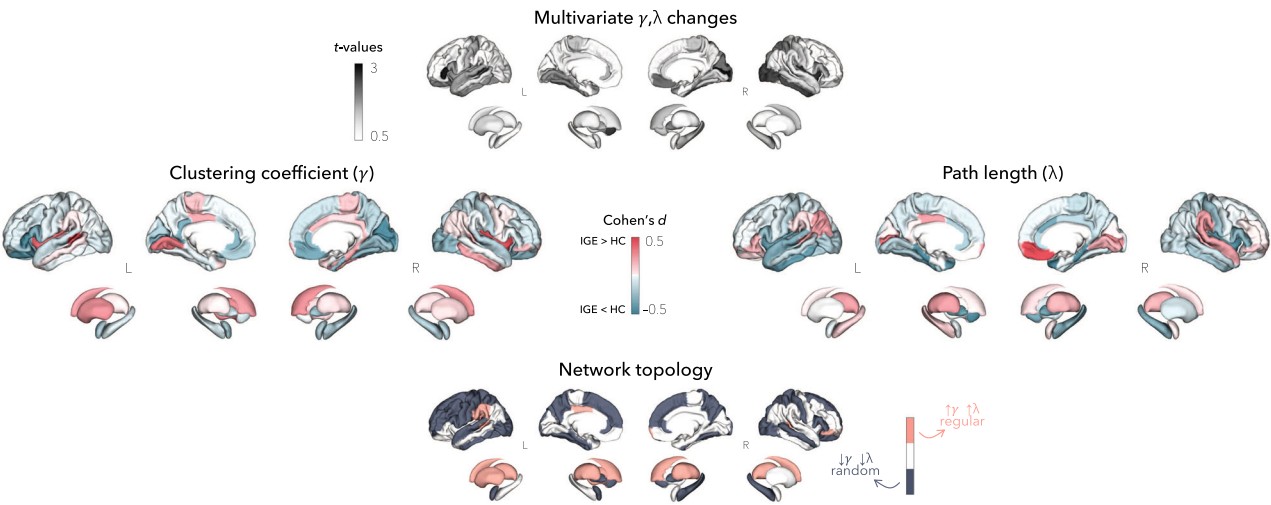

**Fig. 2 Nodal network alterations. a** Graph theoretical analysis of structural covariance between individuals with TLE and controls revealed increased clustering and path length in bilateral orbitofrontal, temporal, and angular cortices, caudate, and putamen, as well as ipsilateral amygdala, revealing a regularized, "lattice-like," arrangement. **b** In IGE, widespread multivariate topological alterations were observed in bilateral fronto-temporo-parietal cortices, right nucleus accumbens, and left pallidum. Clustering and path length effect sizes in these regions suggest a randomized network configuration (decreased clustering and path length). HC = healthy control, IGE = idiopathic generalized epilepsy, TLE = temporal lobe epilepsy.

0.0032) and focal epilepsy ($r = 0.27$, $p_{spin/rand} = 0.015/0.034$). Moreover, in both TLE and IGE, network-level findings did not correlate with any other disease-related transcriptomic maps (range $r = -0.085$–$0.11$, all $p_{spin/rand} > 0.18/0.27$; Fig. 5), with the sole exception of IGE with major depressive disorder ($r = 0.19$, $p_{spin} = 0.015$; this correlation, however, did not survive comparison against randomly selected genes, $p_{rand} = 0.18$).

To ensure that variations in the density of different cell types did not drive transcriptional differences[53], we evaluated whether our epilepsy-specific gene sets were balanced in terms of their cell-type specificity. We separately calculated average cell-type specificity for 29 transcriptomically distinct cell types for hippocampal sclerosis and generalized epilepsy, based on cell-type specificity estimates derived from 17,093 single-nuclei RNA sequencing (snRNA-seq) samples from the dorsolateral prefrontal cortex of three adult human brains[54,55]. In both cases, preponderance of cell types was assessed against null distributions with identical number of genes (see Materials and Methods) and

showed no significant differences in their cell-type specificity (hippocampal sclerosis: $p_{rand} > 0.19$, generalized epilepsy: $p_{rand} > 0.11$). Moreover, average cell-type specificity in hippocampal sclerosis and generalized epilepsy gene sets were overall more similar to each other than to null models (all cell-types $p_{rand} > 0.065$), with the exception of Endo ($p_{rand} < 0.05$) and ExN1 ($p_{rand} < 0.05$).

**Associations with standard neuroimaging parameters and clinical variables.** Compared with univariate mapping of cortical thickness and subcortical volume changes, structural covariance specifically addresses inter-regional structural network organization in TLE and IGE. To evaluate whether TLE-related alterations in covariance patterns are explainable by regional atrophy alone[56], we first compared atrophy profiles in patients relative to controls using surface-based linear models[3]. Patterns of atrophy in TLE and IGE were then spatially compared to multivariate (combined clustering

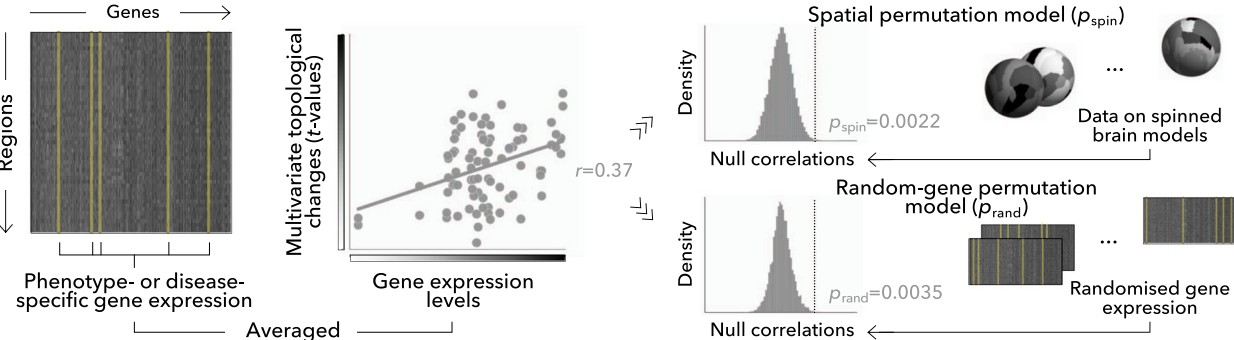

**a** | Statistical testing of imaging-transcriptomics associations

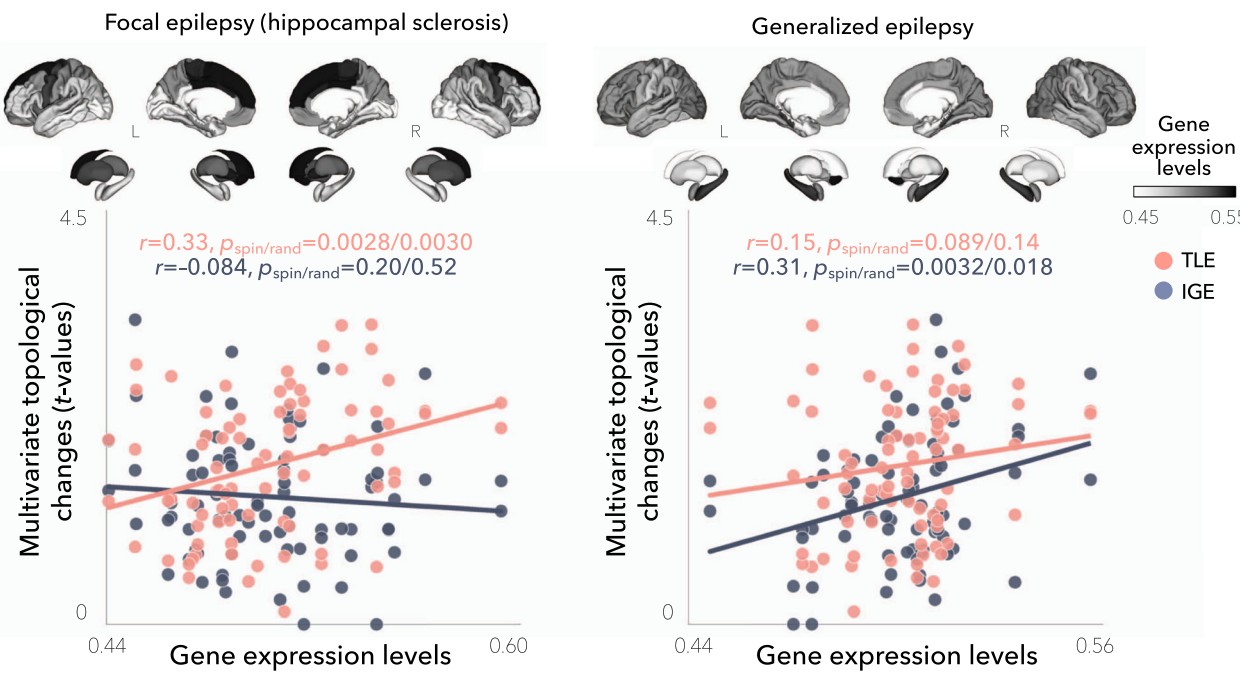

**b** | Transcriptomic associations with TLE and IGE epilepsy risk genes

**Fig. 3 Imaging-transcriptomic associations. a** Schematic of the approaches for statistical testing of imaging-transcriptomic associations. Gene expression data for a subset of phenotype- or disease-specific genes are averaged and spatially compared to the patterns of multivariate topological changes in TLE and IGE independently. Spatial correlations are statistically assessed using one-tailed, non-parametric tests: (i) spatial permutation models, which preserve the spatial autocorrelation of brain maps ($p_{spin}$; 10,000 permutations), and (ii) permutation models, which generate null distributions from randomised gene expression data with identical length as the original gene set ($p_{rand}$; 10,000 permutations). **b** Gene expression levels associated with two distinct epilepsy subtypes (focal epilepsy with hippocampal sclerosis and generalized epilepsy) were mapped to cortical and subcortical surface templates and spatially compared to patterns of multivariate topological alterations (which combined clustering and path length; see Fig. 2) across cortical and subcortical regions ($n = 82$) using one-tailed, non-parametric tests. In TLE, spatial associations between microarray data and multivariate topological changes were strongest for expression levels of hippocampal sclerosis genes ($r = 0.33$, $p_{spin} = 0.0028$). On the other hand, in IGE, spatial associations were strongest for expression levels of generalized epilepsy genes ($r = 0.31$, $p_{spin} = 0.0032$). Both TLE- and IGE-specific imaging-transcriptomic associations were robust against null distributions of effects based on selecting random genes from the full gene set (TLE: $p_{rand} = 0.0030$, IGE: $p_{rand} = 0.018$). HC = healthy control, IGE = idiopathic generalized epilepsy, TLE = temporal lobe epilepsy, $p_{spin}$ = $p$-value corrected against a null distribution of effects using a spatial permutation model, $p_{rand}$ = $p$-value corrected against a null distribution of effects using a "random-gene" permutation model.

and path length) covariance network changes and statistically assessed via non-parametric spin tests[37]. As in previous studies[3,39], patients with TLE showed profound atrophy in bilateral superior parietal (left/right $p_{FDR} = 2.86 \times 10^{-29}/4.50 \times 10^{-27}$), precuneus (left/right $p_{FDR} = 3.54 \times 10^{-29}/3.32 \times 10^{-22}$), precentral (left/right $p_{FDR} = 3.95 \times 10^{-21}/3.12 \times 10^{-20}$), and paracentral (left/right $p_{FDR} = 1.75 \times 10^{-19}/5.41 \times 10^{-18}$) cortices, as well as ipsilateral hippocampus ($p_{FDR} = 2.32 \times 10^{-186}$) and thalamus ($p_{FDR} = 1.25 \times 10^{-67}$; Supplementary Fig. 2a). In contrast, patients with IGE

showed predominant atrophy in bilateral precentral cortices (left/right $p_{FDR} = 2.94 \times 10^{-14}/7.75 \times 10^{-12}$) and thalamus (left/right $p_{FDR} = 8.63 \times 10^{-14}/1.70 \times 10^{-14}$; Supplementary Fig. 2b). The spatial pattern of multivariate topological changes, however, did not closely correspond to areas of atrophy in TLE ($r = 0.097$, $p_{spin} = 0.21$) nor IGE ($r = -0.058$, $p_{spin} = 0.31$), suggesting that covariance changes may not be fully explainable by the spatial distributions of cortical thickness and subcortical volume changes in the same conditions. Moreover, imaging-transcriptomics associations were

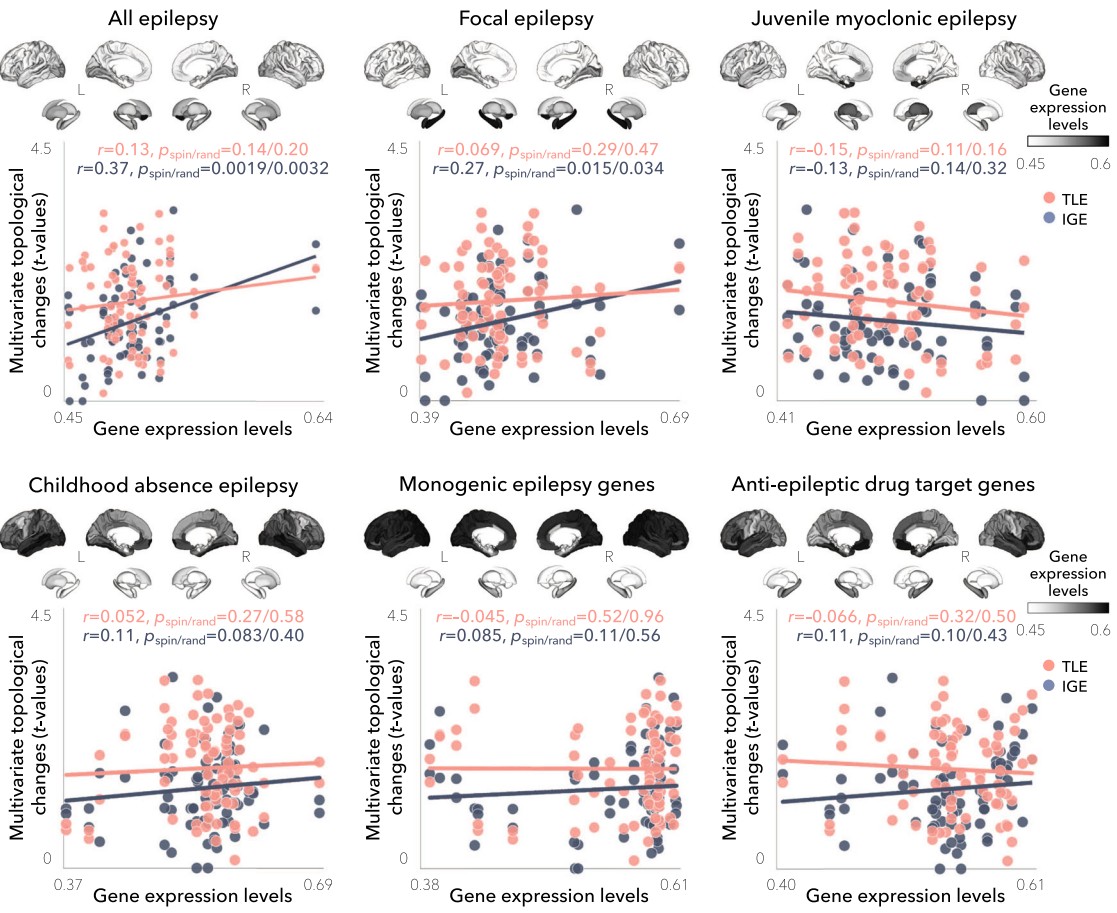

**Fig. 4 Relations between epilepsy gene expression and network topology.** Gene expression levels associated with (i) all other epilepsy subtypes (all epilepsy, focal epilepsy, juvenile myoclonic epilepsy, and childhood absence epilepsy), (ii) monogenic epilepsy, and (iii) anti-epileptic drug targets were mapped to cortical and subcortical surface templates. Spatial correlations were performed between each of these transcriptomic maps and the patterns of multivariate topological alterations in TLE and IGE across cortical and subcortical regions ($n = 82$) and were statistically assessed using one-tailed, non-parametric tests. In IGE, spatial associations between microarray data and multivariate topological changes were significant for expression levels of all epilepsy genes ($r = 0.37$, $p_{spin} = 0.0019$) and focal epilepsy ($r = 0.27$, $p_{spin} = 0.015$). In TLE, network associations did not correlate with any other epilepsy-related transcriptomic maps. HC = healthy control, IGE = idiopathic generalized epilepsy, TLE = temporal lobe epilepsy, $p_{spin}$ = p-value corrected against a null distribution of effects using a spatial permutation model, $p_{rand}$ = p-value corrected against a null distribution of effects using a "random-gene" permutation model.

significantly weaker when derived from regional atrophy patterns (as opposed to multivariate topological changes) in TLE (correlation with gene expression levels of hippocampal sclerosis: $r = 0.041$, $p_{spin/rand} = 0.50/0.91$; Supplementary Fig. 2c) and in IGE (correlation with gene expression levels of generalized epilepsy: $r = -0.083$, $p_{spin/rand} = 0.25/0.83$; Supplementary Fig. 2d).

As seizure focus laterality may differentially affect structural covariance networks[24], we repeated the above analyses in left and right TLE separately, comparing patient subgroups both to controls and to each other. Global increases in clustering and decreases in path length were observed in both left (clustering: $p_{FDR} < 0.05$ at $K = 0.05–0.26$, 28–29; path length: $p_{FDR} < 0.05$ at $K = 0.05$, 0.07–0.22, 25–39; Supplementary Fig. 3a) and right (clustering: $p_{FDR} < 0.05$ at $K = 0.05–0.25$; path length: $p_{uncorr} < 0.05$ at $K = 0.14–0.17$; Supplementary Fig. 3b) TLE patients relative to controls. Similarly, dominant patterns of multivariate (clustering coefficient and path length) topological changes in bilateral lateral occipital cortex ($p_{FDR} < 0.01$), parahippocampus ($p_{FDR} < 0.005$), entorhinal cortex ($p_{FDR} < 0.01$), and insula ($p_{uncorr} < 0.05$), ipsilateral precuneus ($p_{FDR} < 0.01$), anterior cingulate cortex ($p_{FDR} < 0.01$), and superior temporal gyrus ($p_{FDR} < 0.05$), as well as contralateral middle temporal gyrus ($p_{FDR} < 0.05$) were observed when comparing left (Supplementary Fig. 3a) and

right (Supplementary Fig. 3b) TLE cohorts separately to controls. Left TLE additionally showed alterations in bilateral paracentral cortex (left/right $p_{FDR} = 7.00 \times 10^{-5}/0.00015$), as well as precentral (left/right $p_{FDR} = 0.0011/0.00041$) and postcentral gyri (left/right $p_{FDR} = 0.0015/0.0094$), while right TLE additionally showed abnormalities in bilateral hippocampi (left/right $p_{FDR} = 0.0035/0.0024$). Direct comparison of left $vs.$ right TLE revealed no significant global ($p_{FDR} > 0.27$; Supplementary Fig. 4a) nor regional ($p_{FDR} > 0.11$; Supplementary Fig. 4b) differences between the two subcohorts. Effect sizes for clustering and path length indicated network regularization of the mesiotemporal and postcentral gyrus subnetwork in left TLE, but widespread cortical regularization in right TLE; these slight differences in regional topological configurations were confirmed in left $vs.$ right TLE comparisons (Supplementary Fig. 4b). Differences in multivariate topological changes between left and right TLE marginally affected their associations with epilepsy- (Supplementary Fig. 5) and disease-related (Supplementary Fig. 6) risk genes; spatial correlation with expression levels of genes previously associated to hippocampal sclerosis was only significant in left, but not right, TLE. Left TLE also showed a significant association to the 'all epilepsy' subtype ($r = 0.25$, $p_{spin/rand} = 0.022/0.032$). Network alterations in right TLE, on the other hand, correlated with

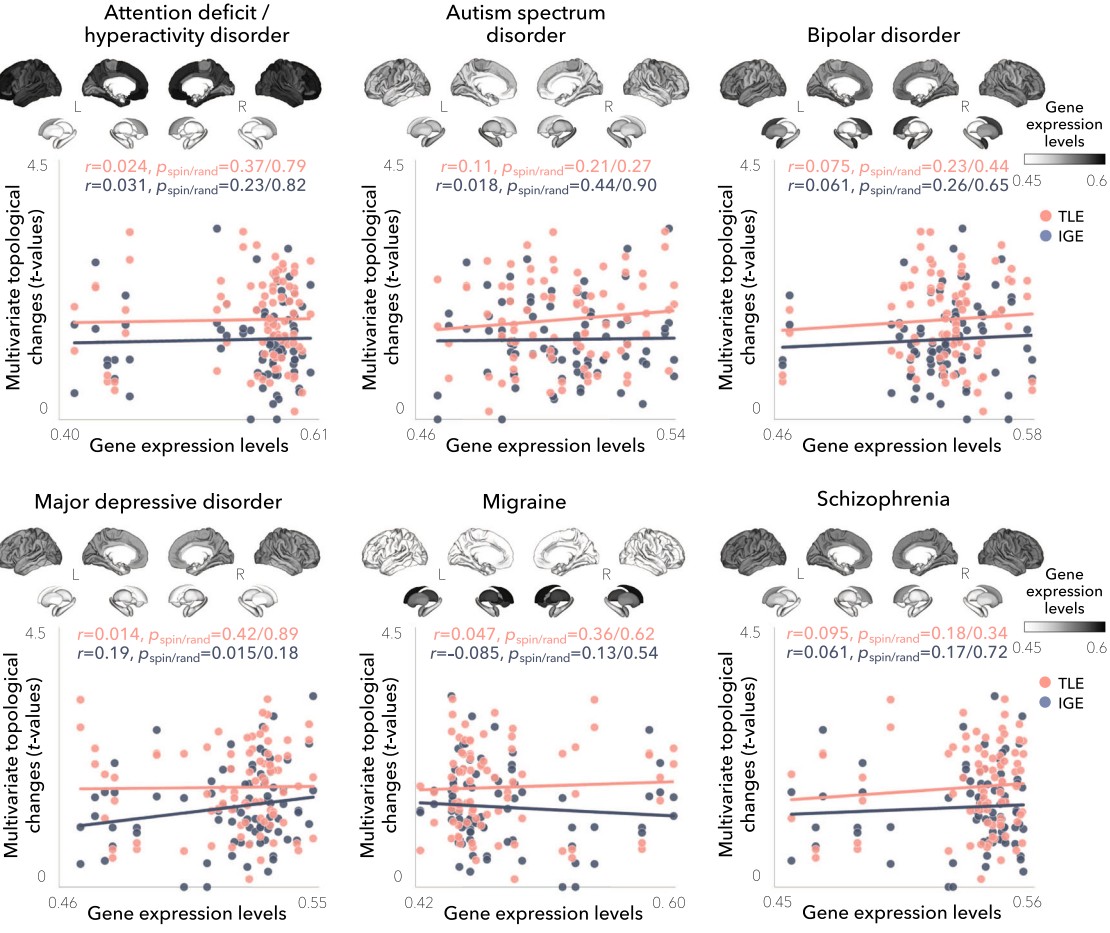

**Fig. 5 Relations between disease-related gene expression and network topology.** Gene expression levels associated with six common neuropsychiatric conditions and/or comorbidities of epilepsy (attention deficit/hyperactivity disorder, autism spectrum disorder, bipolar disorder, major depressive disorder, migraine, and schizophrenia) were mapped to cortical and subcortical surface templates. Spatial correlations were performed between each of these transcriptomic maps and the patterns of multivariate topological alterations in TLE and IGE across cortical and subcortical regions ($n = 82$) and were statistically assessed using one-tailed, non-parametric tests. In IGE, a spatial association between microarray data and multivariate topological changes was significant for expression levels of major depression disorder genes ($r = 0.19$, $p_{spin} = 0.015$). This association, however, did not survive correction against a null distribution of effects based on selecting random genes ($p_{rand} = 0.18$). In TLE, network associations did not correlate with any other disease-related transcriptomic maps. HC = healthy control, IGE = idiopathic generalized epilepsy, TLE = temporal lobe epilepsy, $p_{spin}$ = p-value corrected against a null distribution of effects using a spatial permutation model, $p_{rand}$ = p-value corrected against a null distribution of effects using a "random-gene" permutation model.

transcriptomic maps derived from generalized epilepsy genes ($r = 0.17$, $p_{spin} = 0.048$) and bipolar disorder ($r = 0.20$, $p_{spin} = 0.018$); these correlations, however, did not survive comparison against randomly selected genes, $p_{rand} > 0.14$.

We repeated the structural covariance analyses in patients grouped by duration of illness using a median split approach (TLE = 20 years, $n_{short\text{-}TLE} = 270$, $n_{long\text{-}TLE} = 275$; IGE = 15 years, $n_{short\text{-}IGE} = 137$, $n_{long\text{-}IGE} = 111$). In TLE and IGE, both patient subgroups (short and long duration) showed similar patterns to the overall between-group differences when compared to controls (TLE: Supplementary Fig. 7a; IGE: Supplementary Fig. 8a). Nevertheless, in TLE, we observed a shift in network regularization from fronto-central and limbic regions (shorter duration) to fronto-temporal and limbic regions (longer duration; Supplementary Fig. 7b). Conversely, in IGE, we observed both network randomization (fronto-central regions) and regularization (fronto-parietal regions) in patients with short and long duration (Supplementary Fig. 8b). Direct comparison of patients with short *vs.* long duration of TLE or IGE revealed no significant global (TLE: $p_{FDR} > 0.5$, Supplementary Fig. 9a; IGE: $p_{FDR} > 0.20$, Supplementary Fig. 9b) nor regional (TLE: $p_{FDR} > 0.072$,

Supplementary Fig. 9a) differences between pairs of subcohorts, with the exception of patients with shorter duration of IGE showing multivariate topological changes in bilateral fronto-limbic areas relative to those with longer duration ($p_{FDR} < 0.05$, Supplementary Fig. 9b).

**Robustness of findings across different sites and analysis thresholds.** Despite some site-to-site variability, syndrome-specific global structural covariance differences were overall consistent across sites and similar to those obtained from the multisite aggregation for both TLE and IGE patients (Fig. 1c). As observed in the multisite findings, site-specific increases in clustering coefficient and path length in TLE were most frequently observed in orbitofrontal, temporal, and angular cortices as well as amygdala (Supplementary Fig. 10a). Similarly, in agreement with the multisite findings, site-specific decreases in clustering and path length in IGE were most consistent in fronto-parietal cortices and hippocampus (Supplementary Fig. 10b).

Our findings were not affected by varying the density of structural covariance networks: Across the range of possible thresholds, we observed high correlations among multivariate

topological brain maps computed from thresholded structural covariance matrices in TLE (95.35% of correlations were below $p_{spin} < 0.1$) and IGE (90.86% of correlations were below $p_{spin} < 0.1$; Supplementary Fig. 11). Moreover, we observed comparable associations between topological abnormalities (computed across the range of thresholds) and gene expression levels, with highest stability (% of correlations were below $p_{spin} < 0.1$) in TLE observed for correlations of topological alterations and risk gene expression for focal epilepsy with hippocampal sclerosis (54.00%) and all epilepsy (34.00%). Conversely, stability in IGE was highest for correlations with expression levels of risk genes for generalized epilepsy (22.00%), all epilepsy (30.00%), focal epilepsy (28.00%), childhood absence epilepsy (40.00%), and monogenic epilepsy (42.00%; Supplementary Fig. 12a). In both TLE and IGE, stability of imaging-transcriptomic correlations in neuropsychiatric conditions were overall rather modest, except for correlations with expression levels of risk genes for bipolar disorder in TLE (74.00%), and major depressive disorder (46.00%) as well as schizophrenia (38.00%) in IGE (Supplementary Fig. 12b).

## Discussion

This multisite ENIGMA study is the largest investigation of structural covariance networks in the common epilepsies and bears robust evidence for syndrome-specific topological disruptions. First, despite showing global increases in small-worldness in TLE as compared to controls, regional alterations in orbito-fronto-temporal regions indicated a shift towards a more regularized, "lattice-like", subnetwork configuration. In contrast, IGE presented with widespread decreases in clustering and path length in fronto-temporo-parietal cortices, indicating a more random topology. These syndrome-specific network-level findings were spatially related to the expression pattern of genetic risk factors associated with hippocampal sclerosis and generalized epilepsy in recent GWAS[33]. Findings were highly consistent across sites and methodologies, corroborating robustness and generalizability. Taken together, our study identifies imaging-transcriptomic signatures in the common epilepsies, which ultimately, may facilitate early diagnosis and lead to the development of new and improved treatment strategies.

We performed graph theoretical analysis on MRI-based cortical thickness and subcortical volume correlations in adults with TLE, adults with IGE, and healthy controls[13,19]. Our covariance analysis extends prior research on atrophy mapping by tapping into the topology of inter-regional structural brain networks and describing the network organization underlying whole-brain pathological interactions in the common epilepsies. Using a multisite approach, we showed that patients with TLE preserved an overall small-world configuration with increased clustering and decreased path length over a wide sparsity range. Upon examination of uni- and multivariate regional changes, however, we found key differences between distinct brain subnetworks. Topological alterations were most marked in a subnetwork comprising orbitofrontal, temporal, and angular cortices, pointing to increased local connectivity (i.e., a more regular configuration) in TLE than in healthy controls. This bilateral topological regularization was observed in both left and right TLE patients, albeit more constrained to fronto-temporal cortices in left TLE, a difference that may be attributable to asymmetrical structural damage or to higher connectivity of the dominant hemisphere[57]. Interestingly, although group-level alterations in the hippocampus were modest, with right TLE patients displaying slightly more severe abnormalities than left TLE, intrinsic hippocampal deafferentation may nevertheless contribute to extrahippocampal reconfigurations, affecting neighbouring regions including orbitofrontal and temporal cortices, as well as the amygdala[58]. Given

the high density of connections from the hippocampus to the rest of the brain[59,60], neuronal loss and deafferentation within limbic structures may cause local excess connectivity and decreased internetwork covariance in remote regions. Such a topological shift may be supported by findings in animal models[61] as well as human diffusion MRI[58,62,63] and functional connectivity distance studies[64], which have highlighted imbalances in short- vs. long-range connections in epilepsy-related pathology. More regularized networks are spatially compact, which may facilitate recurrent excitatory activity and high frequency oscillations, and may be attributable to a loss of temporo-limbic structural connections[61,65]. In prior EEG/intracranial EEG studies, network regularization has been reported at seizure onset, a configuration that shifts toward a globally integrated process as the seizure spreads, eventually reaching a random configuration upon seizure termination[66]. Understanding such structural reorganization offers a comprehensive knowledge of the neural substrates and pathophysiological mechanisms of TLE.

TLE is a complex condition that is associated with both atypical early neurodevelopment as well as deviations from typical brain aging processes. Growing evidence supports atypical brain development as a potential etiological factor for TLE, with neuroimaging data revealing quantitative changes in cortical folding, hippocampal malrotations, cortical interface blurring, and connectivity alterations in temporo-limbic networks, which may reflect consequences of malformative processes during prenatal stages[67,68]. Several histological findings also point to atypical tempero-limbic network formation and maturation[63,69]. Following the lifespan, several studies have also suggested interactions between TLE and brain aging, with cross-sectional and longitudinal mesiotemporal volumetry and cortical thickness analysis showing appearance of accelerated brain aging in patients relative to controls[3,70–72]. While often attributed to secondary effects of seizures, these observations may reflect a complex combination of seizure burden, medication load, and psychosocial challenges that medically-intractable patients often face. Notably, when split into short vs. long duration groups, patients with a longer disease duration also presented with topological regularization primarily in temporo-parietal cortices. This is in line with a previous longitudinal structural covariance analysis in a single centre TLE cohort, which suggests that network alterations may intensify over time[24].

In contrast with TLE, overall structural covariance network configurations in IGE showed a tendency away from aberrant local connectivity and towards a more random architecture. On the whole, an increasingly random structural network organization denotes reduced local efficiency but increased global efficiency[22,73]. These global topological findings were complemented by region-specific mapping of graph-theoretical parameters, which also identified widespread regional alterations in IGE patients. Although the pattern was overall mixed across regions—a fortiori in smaller patient subgroups split according to disease duration—showing increases and decreases in both path length and clustering, a large subnetwork comprising frontal, temporal and parietal cortices showed concomitant reductions in path length as well as clustering. A decrease in clustering implies reductions in local specialization, but a decrease in path length (i.e., increased global network efficiency) may indicate an imbalance in the integration and segregation of structural covariance network organization. Such imbalance might explain, at least in part, the ability of seizures to rapidly spread, not just locally but in a diffuse manner within bilateral fronto-temporo-parietal cortices in IGE patients[74]. In rodent models of IGE, fronto-parietal cortices have typically shown increased simultaneous neuronal activity during generalized seizures[75,76]. Moreover, prior neuroimaging work in IGE patients

that analyzed cortical morphology has shown widespread cortical structural network compromise[6,77], with midline frontal and paracentral regions emerging as potential epicenters of morphological abnormalities in IGE[3,78]. Notably, IGE patients also presented with focal patches of network randomization, similar to TLE patients in paralimbic subnetworks. Affected regions included paralimbic cortices, but coupled with subcortical structures, notably the thalamus. Extensive evidence supports atypical thalamo-cortical interactions as being at the core of the pathophysiological network of IGE[7,10,11,39,79], with aberrant thalamo-cortical loops contributing to the generation of spike and slow wave discharges[80]. Alterations of thalamic morphology and metabolism, as well as of its functional and structural connectivity with widespread cortical networks, have also been reported in a convergent neuroimaging literature across several IGE syndromes[10,81,82]. In future work, it will be of interest to explore how consistent, or variable, these topological imbalances in thalamic as well as cortical subnetworks are across different IGE subsyndromes. It is also important to understand effects of clinically relevant parameters, including levels of response to antiseizure medication. In that context, we recommend further increasing the spatial resolution, allowing for a fine-grained assessment of both cortical network architecture and thalamic subdivisions. This could be achieved, for example, by adopting recent approaches that reported structural, functional, and microcircuit anomalies in IGE compared to both TLE patients and healthy controls[6,7].

Connectome topology has been extensively studied in healthy and diseased brains, however, research investigating associations between macroscale findings and the genetic architecture of epilepsy is still in infancy. A recent genome-wide mega-analysis performed in the common epilepsies identified 21 biological candidate genes across 16 risk loci, thus providing initial evidence for epilepsy-associated gene expression changes[33]. By integrating neuroimaging and transcriptional atlas data, here we tested the hypothesis that transcriptomic vulnerability would covary with structural network abnormalities in TLE and IGE. We showed that epilepsy-related variations in brain network topology spatially converged with gene expression profiles of risk genes for each syndrome. Specificity of these associations was supported by the fact that topological alterations did not correlate with transcriptional signatures of several common psychiatric disorders. In the long term, these imaging-transcriptomic associations may form the foundation for translation and clinical studies aiming to tailor therapeutic approaches to specific epilepsy syndromes. From a clinical standpoint, these findings represent a glance of the different pathophysiological anomalies in temporal lobe and generalized epilepsies, and may lead to possible imaging-transcriptomic applications for improved patient stratification. For instance, the transcriptomics-associated network maps identified herein may increase diagnostic sensitivity in both TLE and IGE, while also pointing to different, syndrome-specific, genetically-mediated etiologies. Alternatively, as these subnetwork alterations were associated with syndrome-related risk genes, our findings could provide a foundation for future research aiming to explore whether targeted assessments of these subnetworks can help to discriminate gene variant carriers vs. non-carriers, thus potentially enhancing diagnostics and treatment calibration.

Limitations of imaging-transcriptomic associations with respect to (i) GWAS-identified genes, (ii) microarray vs. RNA-Seq transcriptomic datasets, and (iii) the mapping of single nucleotide polymorphism (SNP) genotyping on gene expression need to be highlighted. Firstly, risk genes used in the current study were obtained from a previously published GWAS from the ILAE Consortium that aggregated data from SNP microarrays

from 15,212 patients with epilepsy and 29,677 controls[33]. The ability of GWAS to identify relevant genes generally scales with overall sample size, and forthcoming studies with larger samples and broader inclusion criteria are expected to expand the catalogue of genes implicated in epilepsy. Rare variants (e.g., causal variants with one rare allele), for instance, are unlikely being tagged by current GWAS-type approaches[83]. Secondly, we derived gene expression from bulk microarray data obtained from six postmortem donor brains from the AHBA, with predominant cortical and subcortical sampling performed in the left hemisphere. By current standards, the AHBA represents a unique and comprehensive resource to associate gene expression and neuroimaging data, offering excellent spatial coverage of nearly the entire human brain and direct mapping of tissue samples to stereotaxic space[53]. On the other hand, while microarray technology remains a popular and cost-effective approach for transcript profiling, it is limited to interrogating only those genes for which probes are designed[84]. Alternative transcriptomics techniques such as RNA-Seq, do not depend on a priori probe selection, and may be more sensitive in identifying genes with low expression and more accurate in detecting expression of common genes[84]. RNA-Seq technology, however, also poses algorithmic and logistical challenges, including the restricted number of samples processed in a single run, elevated costs, data storage requirements, and the absence of an analytical gold standard[85]. Finally, relating GWAS findings to gene expression is a complex process; indeed, GWAS-identified SNPs may occur in non-protein-coding regions[86], may not always affect transcription of the closest gene, and may implicate genes that are located up to 2 Mbps away[87]. SNPs may also influence several steps of gene expression, particularly messenger RNA (mRNA) splicing, stability, and translation, with their precise functional impact on gene function not being fully understood[88,89]. Replication of our findings in more comprehensive, RNA-Seq gene expression datasets may hold significant promise for stratification and effective treatment that can be targeted to the individual patients based on their genetic profile. Once the barriers to widespread use of RNA-Seq are overcome, our understanding of the genetic architecture of the epilepsies will significantly evolve, with the reported risk genes likely being expanded and refined as more genomic and transcriptomic data become available. Despite current limitations of GWAS-identified genes, microarray datasets, SNP genotyping, and gene expression, our findings suggest that genes previously associated with specific epilepsy syndromes were over-represented in regions that share similar topological alterations. In keeping with prior molecular studies in epilepsy[90–93], we speculate that differentially expressed epilepsy-related gene sets may contribute to a selective vulnerability of networks for structural reconfigurations in TLE and IGE. Notably, these distinctive imaging-transcriptomic associations were robust and remained significant after comparison against null distributions derived from randomly selected gene sets of equal length. Nonetheless, differences in the number of genes in each gene lists may have contributed to variability in gene expression profiles. Exploiting individualized gene expression profiles in the same cohort of patients, therefore, seems to be the logical next step to improve imaging-transcriptomic associations and update our understanding of causes and consequences of epilepsy.

Several sensitivity analyses suggested that our findings were not affected by differences in scanners or sites or methodological choices. Site and scanner effects were mitigated for the most part using CovBat, a post-acquisition statistical batch normalization process used to harmonize between-site and between-protocol effects in mean, variance, and covariance, while protecting biological covariates (e.g., disease status)[41]. Multivariate topological findings as well as associations between network-level findings

and gene expression maps were consistently observed across different matrix thresholds. Despite some site-to-site variability in global and regional graph theoretical metrics, findings were overall similar across independent centers and reflected those from the multisite aggregation. As data sharing practices can at times be challenging, in part due to privacy and regulatory protection, ENIGMA represents a practical alternative for standardized data processing and anonymized derivative data[12,94–98]. Notably, the Desikan-Killiany atlas is widely adopted across ENIGMA Working Groups, thus allowing for comparison of results across initiatives. On the other hand, this parcellation is limited by its relatively coarse granularity (68 cortical regions) and variable parcel sizes. In future studies, replication of our findings with higher-resolution cortical and subcortical parcellations that offer better uniformity in areal definition may help to increase generalizability and specificity. Nevertheless, this collaborative effort allowed us to identify an association of brain structural network changes with patterns of expression of genetic risk factors in the common epilepsies, while addressing robustness of effects across clinical subgroups, international sites, and methodological variations. The imaging-transcriptomic associations identified herein could guide diagnosis of common epilepsies, and ultimately, contribute to the development of tailored, individualized, and syndrome-specific therapeutic approaches.

## Methods

**ENIGMA participants.** Epilepsy specialists at each center diagnosed patients according to the seizure and syndrome classifications of the ILAE[99]. Inclusion of adults with TLE was based on the combination of electroclinical features and MRI findings typically associated with underlying hippocampal sclerosis. Inclusion of adults with IGE was based on the presence of tonic-clonic, absence, or myoclonic seizures with generalized spike-wave discharges on EEG. We excluded participants with a progressive or neurodegenerative disease (e.g., Rasmussen's encephalitis, progressive myoclonus epilepsy), malformations of cortical development, tumors, or prior neurosurgery. Healthy controls had no history of mental disorders and were statistically matched for age and sex to the epilepsy subgroups at each site. Local institutional review boards and ethics committees approved each included cohort study, and written informed consent was provided according to local requirements (Table S3).

**Cortical thickness and subcortical volume data.** All participants underwent structural T1-weighted brain MRI scans at each of the 18 participating centers, with scanner descriptions and acquisition protocols detailed elsewhere[39]. Images were independently processed by each center using the standard ENIGMA workflow. In brief, models of cortical and subcortical surface morphology were generated with FreeSurfer 5.3.0[100]. Based on the Desikan-Killiany anatomical atlas[40], cortical thickness was measured across 68 grey matter brain regions and volumetric measures were obtained from 12 subcortical grey matter regions (bilateral amygdala, caudate, nucleus accumbens, pallidum, putamen, thalamus) as well as bilateral hippocampus. Missing cortical thickness and subcortical volume data were imputed with the mean value for that given region; participants with missing data in at least half of the cortical or subcortical brain measures were excluded.

Data were harmonized across scanners and sites using CovBat—a batch-effect correction tool that uses a Bayesian framework to improve the stability of the parameter estimates[41]. Cortical thickness and volumetric measures were corrected for age and sex. Residualized data were z-scored relative to site-matched pooled controls and sorted into measures that were ipsilateral/contralateral to the focus.

**Covariance networks.** Covariance networks were computed from cortical thickness and subcortical volume correlations. Inter-regional association matrices were first generated for each group (TLE, HC$_{TLE}$, IGE, HC$_{IGE}$) and each site with at least 10 participants per diagnostic group ($n_{TLE/HC} = 14$ sites, $n_{IGE/HC} = 10$ sites), resulting in a total of 48 covariance matrices ($R$). In each matrix $R$, an individual entry $R_{i,j}$ (with regions $i$ and $j$) contained the pairwise linear product-moment cross-correlation coefficient of structural morphometry across group- and site-specific subjects.

**Network thresholding.** Prior to analysis, negative correlations were set to zero and covariance network matrices were thresholded (density range of $K = 0.05$–$0.50$, density interval of 0.01). This approach ensured that networks in all groups had an identical number of edges[101] and that group differences were not primarily driven by low-level correlations[102].

Among the network density levels of $K = 0.05$–$0.50$, the network connectedness criterion (≥75% of nodes remain connected to other nodes within the network in at least 90% of sites) was satisfied only in the narrower $K = 0.08$–$0.50$ range. For the main regional topological analyses, structural networks were constructed at a density of $K = 0.08$.

**Strength of cortical thickness and subcortical volume correlations.** We compared the mean strength of positive correlations (computed from the site-specific thresholded structural covariance networks) in patients relative to controls using Student's t-tests. Statistical differences were evaluated at every density within the $K = 0.08$–$0.50$ range.

**Global network properties.** From the thresholded structural covariance networks, we computed three global metrics using standard formulas[42,103]: (i) mean clustering coefficient, which quantifies the tendency for brain regions to be locally interconnected with neighboring regions, (ii) mean path length, which quantifies the mean minimum number of edges (i.e., connection between two regions) that separate any two regions in the network, and (iii) small-world index (mean clustering coefficient divided by mean path length), which quantifies both local and global properties. These two metrics, along with their combination (i.e., small-world index), are the most widely used graph theoretical parameters to describe the topology of complex networks. Each measure was normalized relative to corresponding measures from 1000 randomly generated networks with similar degree and weight properties, and subsequently averaged across all cortical and subcortical regions, separately.

**Regional network properties.** Regional differences in topological parameters (clustering coefficient, path length, and small-world index) were assessed using an approach similar to the global network analysis; from the thresholded structural covariance networks, normalized clustering coefficient, normalized path length metrics, as well as their ratio, were computed for every cortical and subcortical brain region. Individual nodal network parameters in patients were compared to controls across sites via Cohen's $d$ effect sizes. From effect size maps, topological profiles were generated to reflect their deviations away from a small-world organization, that is, either network regularization (areas of increased clustering coefficient and path length) or randomization (areas of decreased clustering coefficient and path length)[43]. To signify an overall load of anomalies, we subsequently compared the aggregate of clustering coefficient and path length differences in patients relative to controls using multivariate surface-based linear models. This approach allowed topological changes to be described in a compact manner, and consequently, enabled spatial associations with brain maps of gene expression (see section on Transcriptomic associations). Moreover, by statistically combining clustering coefficient and path length, we leveraged their covariance to obtain a substantial gain in sensitivity, and thus, unveiled subthreshold network properties not readily identified in a single graph theoretical metric.

**Transcriptomic associations.** The Allen Institute for Brain Science released the AHBA—a brain-wide transcriptomic atlas based on microarray expression data from over 20,000 genes sampled across 3702 spatially distinct tissue samples collected from six neurotypical adult whole brains (three Caucasian males, two African American males, and one Caucasian woman)[35]. Microarray expression data were first generated using abagen[104], a toolbox that provides reproducible workflows for processing and preparing gene expression data according to previously established recommendations[53]. Preprocessing steps included intensity-based filtering of microarray probes, selection of a representative probe for each gene across both hemispheres, matching of microarray samples to brain parcels from the Desikan–Killiany atlas[40], within-donor normalization (using the robust sigmoid function and rescaled to the unit interval with the Min-Max function) to mitigate donor-specific effects, and aggregation within parcels and across donors. To account for known differences in microarray expression between broad structural compartments (e.g., cortex vs. subcortex)[105], normalization procedures were performed separately for cortical and subcortical structures[53]. Genes whose similarity across donors fell below a threshold ($r < 0.2$) were removed, leaving a total of 12,668 genes for analysis. All gene sets were mapped to cortical and subcortical regions using the Allen Human Brain Atlas[35] and projected to surface templates.

Leveraging a recently published GWAS from the International League Against Epilepsy Consortium on Complex Epilepsies[33], we extracted the most likely genes associated with significant genome-wide loci in the common epilepsies. Briefly, samples from the ILAE Consortium cohort and population-based datasets were genotyped on SNP arrays and quality-controlled[106,107]. Functional mapping and annotation (FUMA) of GWAS[108] was used to map genome-wide significant loci of all epilepsy phenotypes to genes in and around these loci, resulting in a total of 146 mapped genes, with some genes being associated with multiple phenotypes. Our main analysis examined associations between MRI-derived network topological changes in TLE and IGE to gene expression levels of focal epilepsy with hippocampal sclerosis ($n_{genes} = 5$) and generalized epilepsy ($n_{genes} = 43$). To assess specificity of our imaging-transcriptomic associations with focal epilepsy with hippocampal sclerosis and generalized epilepsy phenotypes, we also performed spatial correlations with every other epilepsy phenotype (all epilepsy: $n_{genes} = 16$,

focal epilepsy: $n_{genes} = 9$, juvenile myoclonic epilepsy: $n_{genes} = 13$, childhood absence epilepsy: $n_{genes} = 4$). Due to the low number of genes in some epilepsy phenotypes, our gene lists included all genes mapped in and around significant genome-wide loci (i.e., regions encompassing all SNPs with $p < 5 \times 10^{-4}$ that were in linkage disequilibrium $= R^2 > 0.2$), that is, no biological prioritization or pre-selection of genes was performed.

Using a similar approach, we also examined associations with transcriptomic maps derived from monogenic epilepsy genes ($n_{genes} = 69$)[45] and genes that are targets of currently used anti-epileptic drugs ($n_{genes} = 44$)[46]. To interrogate disease specificity, we also queried additional lists of disease-related genes (obtained from other recently published GWAS), including gene sets for attention deficit/ hyperactivity disorder ($n_{genes} = 15$)[47], autism spectrum disorder ($n_{genes} = 19$)[48], bipolar disorder ($n_{genes} = 25$)[49], major depressive disorder ($n_{genes} = 152$)[50], migraine ($n_{genes} = 22$)[51], and schizophrenia ($n_{genes} = 110$)[52].

To evaluate whether interregional differences in postmortem gene expression were driven by varying densities of different cell types, we obtained previously published cell type gene expression results sampled in the adult human dorsolateral prefrontal cortex[54]. Adopting this approach, specificity was assessed in 29 single-nucleus subtypes: Astro1, Astro2, Astro3, Astro4, Endo, ExN1, ExN2a, ExN2b, ExN3e, ExN4, ExN5b, ExN6a, ExN6b, ExN8, InN1a, InN1b, InN1c, InN3, InN4a, InN4b, InN6a, InN6b, InN7, InN8, Microglia, Oligo, OPC1, OPC2, and VSMC. For each TLE- and IGE-related gene set (i.e., hippocampal sclerosis and generalized epilepsy risk genes), we calculated the averaged specificity for each of the 29 cell types. To statistically assess whether different cell types were overexpressed in each epilepsy gene list, we compared their average specificity to null distributions. For each cell type, a null distribution was generated by randomly selecting genes (10,000 iterations) and averaging cell type specificity scores. For consistency, the number of random genes selected was identical to the number of genes in the epilepsy-specific gene set. The empirical (i.e., original) specificity score was compared against the null distribution determined by the ensemble of randomly calculated specificity scores. Using this approach, we also directly compared expression of cell types in hippocampal sclerosis and generalized epilepsy gene sets by comparing the difference in their average specificities (i.e., hippocampal sclerosis—generalized epilepsy) to null distributions [composed of (i) hippocampal sclerosis—randomly selected genes and (ii) generalized epilepsy—randomly selected genes]. Empirical and randomly generated gene lists whose averaged scores equaled zero were discarded from the analysis.

### Non-parametric tests
*Spatial permutation models.* The intrinsic smoothness in two given brain maps may inflate the significance of their spatial correlation, if the spatial dependencies in the data are not taken into account[37]. Statistical significance of spatial correlations (e.g., between multivariate topological patterns and transcriptomic maps) was assessed using spin permutation tests[44]. Here we used a previously published framework for parcellated data[109] and adapted it to include subcortical structures. Briefly, we generated a surface model of all 14 subcortical structures and merged it with the initial reconstructed FreeSurfer cortical surface model. The resulting surface mesh thus comprised adequately placed cortical and subcortical vertices, with original volumetric distances being preserved. We inflated this new combined cortical/ subcortical surface model and mapped its vertices to a sphere. In doing so, we obtained the spherical coordinates of each of our 68 cortical regions and 14 sub-cortical structures[110]. We then applied randomly sampled rotations (10,000 repetitions unless specified otherwise) about three axes (x: left-right, y: rostral-caudal, z: dorsal-ventral) at three randomly generated angles, $\theta x$, $\theta y$, and $\theta z \in [0, 2\pi]$[109]. Following sphere rotation, coordinates of the rotated regions were matched to coordinates of the original regions using Euclidean distance. This matching yielded a mapping from the set of regions to itself, thus allowing the assignment of original values to rotated regions. The empirical correlation coefficients are then compared against the null distribution determined by the ensemble of spatially permuted correlation coefficients. Given that spin permutation models have been pre-dominantly constrained to cortical brain maps, we cross-validated our combined cortical/subcortical spin permutation model to a previously published variogram-matching model[111,112]. This method generates surrogate brain maps with matched spatial autocorrelation to that of a target brain, and has been applied to subcortical structures. As recommended[111], surrogate maps were generated using surface-based geodesic distance between cortical regions and three-dimensional Euclidean distance between subcortical and cortical/subcortical regions. Variogram-matching null distributions were then generated from randomly shuffling surrogate maps while preserving the distance-dependent correlation between elements of the brain map. As illustrated in Supplementary Fig. 13, null distributions generated from the spin method were in close agreement with the variogram-matching model and provided nearly identical p-values for spatial correlations, thus supporting the validity of our combined cortical/subcortical spin permutation method.

*Random-gene permutation models.* Due to differences in the length of the various gene sets (range $n_{genes} = 5$–152), we also assessed our imaging-transcriptomic associations against "random-gene" null distributions[38]. Null distributions were generated by correlating multivariate topological alterations in patients to tran-scriptomic maps derived from randomly selecting gene sets of equal length from the pool of all $n = 12,668$ genes (10,000 iterations). This approach thus allowed us to examine whether our empirical correlation coefficients are significantly larger than those derived from null distributions of identically sized, and randomized, gene sets.

**Associations with clinical variables**. As seizure focus lateralization may differ-entially impact topological organization of structural covariance networks[25], we repeated the graph theoretical and transcriptomics analyses by comparing (i) left ($n_{LTLE} = 321$) and right ($n_{RTLE} = 257$) TLE independently to controls and (ii) directly comparing left vs. right TLE. Imaging-transcriptomic correlations assessing the spatial overlap between network alterations and epilepsy- and disease-related gene expression levels were assessed in left and right TLE independently.

To study the effects of duration of illness on structural covariance networks, we repeated the graph theoretical analyses by comparing patients with short duration of TLE or IGE (TLE duration <20 years, $n_{short-TLE} = 270$; IGE duration <15 years, $n_{short-IGE} = 137$) and patients with long duration of TLE or IGE (TLE duration ≥20 years, $n_{long-TLE} = 275$; IGE duration ≥15 years, $n_{long-IGE} = 111$) independently to controls, (ii) directly comparing short vs. long TLE, and (iii) directly comparing short vs. long IGE. Median splits were used to group short vs. long duration patients.

### Reproducibility and sensitivity analyses
*Reproducibility across sites.* To address reproducibility of our findings across dif-ferent sites, we repeated our multivariate topological analysis independently in each site ($n_{TLE/HC} = 14$ sites, $n_{IGE/HC} = 10$ sites).

*Stability across matrix thresholds.* To verify that results were not biased by choosing a particular threshold, we repeated the network analyses and associations with disease-related gene expression levels across the range of matrix thresholds ($K = 0.05$–0.50 with increments of 0.01). Specifically, Hotelling's $T^2$ multivariate (clustering coefficient and path length) topological changes comparing patients to controls were computed from structural covariance networks thresholded at every density and pairwise spatial correlations between all pairs of multivariate brain maps were performed. Spatial correlations between density-specific multivariate topological alterations and all epilepsy- and disease-related transcriptomic maps were also assessed. Significance testing of these correlations was assessed via spin permutation tests with 1,000 repetitions.

**Reporting summary**. Further information on research design is available in the Nature Research Reporting Summary linked to this article.

## Data availability
Neuroimaging (ENIGMA-Epilepsy meta-analysis of summary statistics[39]) and transcriptomic data[35] related to this paper are freely available for download (https://github.com/MICA-MNI/ENIGMA). Requests for subject-level neuroimaging data can be proposed to the ENIGMA-Epilepsy Working Group (http://enigma.ini.usc.edu/).

## Code availability
All codes needed to analyze the data is openly available in the ENIGMA Toolbox (https://github.com/MICA-MNI/ENIGMA)[110] and is complemented with an expandable online documentation (https://enigma-toolbox.readthedocs.io/).

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

## Acknowledgements
The authors would like to express their gratitude to the open science initiatives that made this work possible: (i) the ENIGMA Consortium (core funding for ENIGMA was provided by the NIH Big Data to Knowledge (BD2K) program under consortium grant U54 EB020403 to P.M.T.) and (ii) The Allen Human Brain Atlas and the abagen toolbox (https://abagen.readthedocs.io/). Sara Larivière was funded by the Fonds de la Recherche du Québec—Santé (FRQ-S), Canadian Institutes of Health Research (CIHR), and the Ann and Richard Sievers Neurosciene Award. Raul Rodríguez-Cruces was funded by the FRQ-S. Jessica Royer was supported by a Canadian Open Neuroscience Platform (CONP) fellowship and CIHR. Gavin P. Winston was funded by the Medical Research Council (G080212 and MR/M00841X/1). Lorenzo Caciagli acknowledges support from a Berkeley Fellowship jointly awarded by UCL and Gonville and Caius College, Cambridge, and by Brain Research UK (award 14181). The UNAM site was funded by UNAM-DGAPA (IB201712, IG200117) and Conacyt (181508 and Programa de Laboratorios Nacionales). Mark Richardson was funded by UK Medical Research Council grant MR/K013998/1. Fernando Cendes and Clarissa Yasuda were supported by the São Paulo Research Foundation (FAPESP), Grant # 2013/07559-3 (BRAINN—Brazilian Institute of Neuroscience and Neurotechnology). Stefano Meletti and Anna Elisabetta Vaudano were supported by the Ministry of Health (MOH), grant # NET-2013-02355313. Paul Thompson was funded by R01 NS106957 and P41 EB015922. Carrie R. McDonald was supported by ENIGMA-R21 (NIH/NINDS R21NS107739) and R01 (NIH/NINDS R01 NS122827). Sanjay M. Sisodiya was supported by the Epilepsy Society, UK. Boris C. Bernhardt acknowledges research support from the National Science and Engineering Research Council of Canada (NSERC Discovery-1304413), the CIHR (FDN-154298, PJT-174995), SickKids Foundation (NI17-039), Azrieli Center for Autism Research (ACAR-TACC), BrainCanada, FRQ-S, and the Tier-2 Canada Research Chairs program.

## Author contributions
Study conceptualization: Sa.L., B.C.B., C.R.M., A.L., M.E.C., N.B., A.B., and S.M.S. Analysis: Sa.L. and B.C.B.; individual study sites represented by other coauthors provided preprocessed MRI data and clinical specifiers. ENIGMA Epilepsy Leadership: C.D.W., S.M.S., C.R.M., and P.M.T. Writing: S.L. and B.C.B.; revised and approved by: J.R., R.R-C., C.P., M.E.C., A.G., Lu.C., S.S.K., F.C., C.L.Y., L.B., E.G., N.K.F., M.D., F.v.P., So.L., C.R., R.W., P.M., Ra.K., T.J.O., B.S., L.V., P.M.D., E.L., A.E.V., S.M., M.T., S.A., C.P.D., G.L.C., N.D., Re.K., G.D.J., M.K., M.M., Mi.S., R.H.T., H.S.Z., E.D-B., J.Z., G.P.W., A.G., A.S., V.K.T., B.A.K.K., M.L., R.G., K.H., S.F., T.R., B.W., C.D., J.A., S.J.A.C., E.A., M.P.R., O.D., Ma.S., P.S., D.T., E.K., S.N.H., S.B.V., Lo.C., J.S.D., C.D.W., P.M.T., S.M.S., A.B., A.L., C.R.M., N.B, and B.C.B.

## Funding
P.M.T. is funded in part by a research grant from Biogen, Inc. for work unrelated to the current manuscript.

## Competing interests
The authors declare no competing interests.

## Additional information

Sara Larivière[1✉], Jessica Royer[1], Raúl Rodríguez-Cruces[1], Casey Paquola[2], Maria Eugenia Caligiuri[3], Antonio Gambardella[3,4], Luis Concha[5], Simon S. Keller[6,7], Fernando Cendes[8], Clarissa L. Yasuda[8], Leonardo Bonilha[9], Ezequiel Gleichgerrcht[10], Niels K. Focke[11], Martin Domin[12], Felix von Podewills[13], Soenke Langner[14], Christian Rummel[15], Roland Wiest[15], Pascal Martin[16], Raviteja Kotikalapudi[16], Terence J. O'Brien[17,18], Benjamin Sinclair[17,18], Lucy Vivash[17,18], Patricia M. Desmond[18], Elaine Lui[18], Anna Elisabetta Vaudano[19,20], Stefano Meletti[19,20], Manuela Tondelli[20,21], Saud Alhusaini[22,23], Colin P. Doherty[24,25], Gianpiero L. Cavalleri[22,25], Norman Delanty[22,25], Reetta Kälviäinen[26,27], Graeme D. Jackson[28], Magdalena Kowalczyk[28], Mario Mascalchi[29], Mira Semmelroch[28], Rhys H. Thomas[30], Hamid Soltanian-Zadeh[31,32], Esmaeil Davoodi-Bojd[33], Junsong Zhang[34], Gavin P. Winston[35,36,37], Aoife Griffin[38], Aditi Singh[38], Vijay K. Tiwari[38], Barbara A. K. Kreilkamp[11], Matteo Lenge[39,40], Renzo Guerrini[39], Khalid Hamandi[41,42], Sonya Foley[42], Theodor Rüber[43,44,45], Bernd Weber[46], Chantal Depondt[47], Julie Absil[48], Sarah J. A. Carr[49], Eugenio Abela[49], Mark P. Richardson[49], Orrin Devinsky[50], Mariasavina Severino[51], Pasquale Striano[51,52], Domenico Tortora[52], Erik Kaestner[53], Sean N. Hatton[54], Sjoerd B. Vos[36,37,55], Lorenzo Caciagli[36,37], John S. Duncan[36,37], Christopher D. Whelan[22], Paul M. Thompson[56], Sanjay M. Sisodiya[36,37], Andrea Bernasconi[57], Angelo Labate[58], Carrie R. McDonald[53], Neda Bernasconi[57] & Boris C. Bernhardt[1✉]

[1]Multimodal Imaging and Connectome Analysis Laboratory, McConnell Brain Imaging Centre, Montreal Neurological Institute and Hospital, McGill University, Montreal, QC, Canada. [2]Institute for Neuroscience and Medicine (INM-1), Forschungszentrum Jülich, Jülich, Germany. [3]Neuroscience Research Center, University Magna Græcia, Catanzaro, CZ, Italy. [4]Institute of Neurology, University Magna Græcia, Catanzaro, CZ, Italy. [5]Institute of Neurobiology, Universidad Nacional Autónoma de México, Querétaro, México. [6]Institute of Systems, Molecular and Integrative Biology, University of Liverpool, Liverpool, UK. [7]Walton Centre NHS Foundation Trust, Liverpool, UK. [8]Department of Neurology, University of Campinas–UNICAMP, Campinas, São Paulo, Brazil. [9]Department of Neurology, Emory University, Atlanta, GA, USA. [10]Department of Neurology, Medical University of South Carolina, Charleston, SC, USA. [11]Department of Neurology, University of Medicine Göttingen, Göttingen, Germany. [12]Institute of Diagnostic Radiology and Neuroradiology, Functional Imaging Unit, University Medicine Greifswald, Greifswald, Germany. [13]Department of Neurology, University Medicine Greifswald, Greifswald, Germany. [14]Institute of Diagnostic Radiology and Neuroradiology, University Medicine Greifswald, Greifswald, Germany. [15]Support Center for Advanced Neuroimaging (SCAN), University Institute of Diagnostic and Interventional Neuroradiology, University Hospital Bern, Bern, Switzerland. [16]Department of Neurology and Epileptology, Hertie Institute for Clinical Brain Research, University of Tübingen, Tübingen, Germany. [17]Department of Neuroscience, Central Clinical School, Alfred Hospital, Monash University, Melbourne, Melbourne, VIC, Australia. [18]Departments of Medicine and Radiology, The Royal Melbourne Hospital, The University of Melbourne, Parkville, VIC, Australia. [19]Neurology Unit, OCB Hospital, Azienda Ospedaliera-Universitaria, Modena, Italy. [20]Department of Biomedical, Metabolic and Neural Science, University of Modena and Reggio Emilia, Modena, Italy. [21]Primary Care Department, Azienda Sanitaria Locale di Modena, Modena, Italy. [22]Department of Molecular and Cellular Therapeutics, The Royal College of Surgeons in Ireland, Dublin, Ireland. [23]Department of Neurology, Yale University School of Medicine, New Haven, CT, USA. [24]Department of Neurology, St James' Hospital, Dublin, Ireland. [25]FutureNeuro SFI Research Centre, Dublin, Ireland. [26]Epilepsy Center, Neuro Center, Kuopio University Hospital, Member of the European Reference Network for Rare and Complex Epilepsies EpiCARE, Kuopio, Finland. [27]Faculty of Health Sciences, School of Medicine, Institute of Clinical Medicine, University of Eastern Finland, Kuopio, Finland. [28]Florey Institute of Neuroscience and Mental Health, University of Melbourne, Melbourne, VIC 3010, Australia. [29]Neuroradiology Research Program, Meyer Children Hospital of Florence, University of Florence, Florence, Italy. [30]Transitional and Clinical Research Institute, Newcastle University, Newcastle upon Tyne, UK. [31]Contol and Intelligent Processing Center of

Excellence (CIPCE), School of Electrical and Computer Engineering, University of Tehran, Tehran, Iran. [32]Departments of Research Administration and Radiology, Henry Ford Health System, Detroit, MI, USA. [33]Department of Neurology, Henry Ford Health System, Detroit, MI, USA. [34]Cognitive Science Department, Xiamen University, Xiamen, China. [35]Division of Neurology, Department of Medicine, Queen's University, Kingston, ON, Canada. [36]Department of Clinical and Experimental Epilepsy, UCL Institute of Neurology, London, UK. [37]Chalfont Centre for Epilepsy, Bucks, UK. [38]Wellcome-Wolfson Institute for Experimental Medicine, School of Medicine, Dentistry & Biomedical Science, Queens University Belfast, Belfast, UK. [39]Child Neurology Unit and Laboratories, Neuroscience Department, Children's Hospital A. Meyer-University of Florence, Florence, Italy. [40]Functional and Epilepsy Neurosurgery Unit, Neurosurgery Department, Children's Hospital A. Meyer-University of Florence, Florence, Italy. [41]The Welsh Epilepsy Unit, Department of Neurology, University Hospital of Whales, Cardiff, UK. [42]Cardiff University Brain Research Imaging Centre (CUBRIC), College of Biomedical Sciences, Cardiff University, Cardiff, UK. [43]Department of Epileptology, University of Bonn Medical Center, Bonn, Germany. [44]Epilepsy Center Frankfurt Rhine-Main, Department of Neurology, Goethe-University Frankfurt, Frankfurt am Main, Germany. [45]Center for Personalized Translational Epilepsy Research (CePTER), Goethe-University Frankfurt, Frankfurt am Main, Germany. [46]Institute of Experimental Epileptology and Cognition Research, University Hospital Bonn, Bonn, Germany. [47]Department of Neurology, Hôpital Erasme, Université Libre de Bruxelles, Brussels, Belgium. [48]Department of Radiology, Hôpital Erasme, Université Libre de Bruxelles, Brussels, Belgium. [49]Division of Neuroscience, Institute of Psychiatry, Psychology and Neuroscience, King's College London, London, UK. [50]Department of Neurology, NYU Grossman School of Medicine, New York, NY, US. [51]IRCCS Istituto Giannina Gaslini, Genova, Italy. [52]Department of Neurosciences, Rehabilitation, Ophthalmology, Genetics, Maternal and Child Health, University of Genova, Genova, Italy. [53]Department of Psychiatry, Center for Multimodal Imaging and Genetics, University of California San Diego, La Jolla, CA, US. [54]Department of Neurosciences, Center for Multimodal Imaging and Genetics, University of California San Diego, La Jolla, CA, US. [55]Centre for Medical Image Computing, University College London, London, UK. [56]Imaging Genetics Center, Mark & Mary Stevens Institute for Neuroimaging and Informatics, Keck School of Medicine, University of Southern California, Los Angeles, CA, US. [57]Neuroimaging of Epilepsy Laboratory, McConnell Brain Imaging Centre, Montreal Neurological Institute and Hospital, McGill University, Montreal, QC, Canada. [58]Neurology, BIOMORF Dipartment, University of Messina, Messina, Italy.
✉email: sara.lariviere@mail.mcgill.ca; boris.bernhardt@mcgill.ca

