## [Peer Review File · Nature Communications]

Structural network alterations in focal and generalized epilepsy assessed in a worldwide ENIGMA study follow axes of epilepsy risk gene expressionREVIEWER COMMENTS

Reviewer #1 (Remarks to the Author):

The manuscript by Larivière and colleagues uses ENIGMA consortium data to investigate structural covariance in epilepsy subtypes, as well as the association of altered structural covariance with patterns of brain gene expression for genes that have previously been implicated in risk for epilepsy. The authors report regionally specific structural covariance alterations that differ between IGE and TLE, and they report that this regional specificity co-varies with patterns of expression of genes associated with TLE and IGE respectively. The proposed article has many strengths. The sample size is large (n=578 with temporal lobe epilepsy[TLE], 288 with idiopathic generalized epilepsy [IGE], 1,328 healthy controls) that is well suited for the questions being asked. The use of the Allen Brain Atlas as a transcriptomic reference is not without weaknesses, but it is consistent with current standards in the field. The methods are generally sound in our opinion, in particular the use of covBat is a nice application of a recent methodological development; the treatment of spatial autocorrelation is rigorous; and the analysis suggesting specificity to the epilepsy gene set as opposed to genes associated with other neuropsychiatric disorders is a nice control. The findings are a valuable contribution to the growing field of imaging-genetics. There are however some methodological and interpretative issues that limit our enthusiasm for the paper in its present form. We hope the following comments/suggestions are helpful.

- The effects of SNPs on gene expression are not always straightforward, and don't always simply affect transcription of the closest gene. The authors should describe how SNPs were mapped to genes for the gene sets they utilize and discuss the limitations of these approaches in the discussion section.

- Structural covariance relates to several other variables, such as transcriptomic similarity, distance, and structural connectivity. The current understanding and ambiguity regarding biological interpretations of structural covariance should be discussed in either the introduction or discussion. Relatedly, the authors should discuss the question of whether structural covariance network differences primarily reflect neurodevelopment deviations that cause epilepsy, or could they be the consequence of recurrent seizures (or even medications) associated with the condition?

- It is known that preponderance of different cell types is a main driver of inter-regional differences in post-mortem gene expression. Is this reflected in differences in the gene sets for TLE and IGE? In other words could the spatial differences in gene expression reflect cell type differences in the gene set?

- Given evidence that network edge weight can be associated with topological differences in some network phenotypes, it would be helpful to know if there were differences in total weight between groups

- Presumably this information is in the ENIGMA toolbox, but the implementation of the spin test requires a few more details to understand what was done. What was the treatment of subcortical structures? How was the regional parcellation dealt with in the rotated null model?

- How do areas of topological differences relate to any areas of case control differences in mean thickness or volume? If they overlapped this would potentially affect the interpretation. See for example a recent study in schizophrenia discussing this issue <https://ajp.psychiatryonline.org/doi/10.1176/appi.ajp.2019.18040380>. This comparison could be descriptive with a quantitative analysis left for future work.

- Ideally another (and more uniform) parcellation would also be tested. Doing this would require a lot of work and we do not think it's necessary to actually perform this additional analysis, but the reliance on the Desikan parcellation should be discussed as a weakness in the discussion.

- Are the disease effects (correspondence between GE and group neuroimaging differences) robust to the exact number of genes nominated as significant for each disease? Since the number of significant GWAS hits is proportional to sample size of studies, this number would be expected to change substantially in the future for all of the disorders considered. It would be interesting to vary the threshold for SNP significance (and thus ultimately the number of genes in the gene set) and see if main results are robust to varying this threshold. This is particularly relevant to the FTE set that only includes 3 genes and might be more susceptible to a false positive result.

Signed,
Ayan Mandal
Aaron Alexander-Bloch

Reviewer #2 (Remarks to the Author):

The manuscript by Lariviere et al. studied changes in brain structure in epileptic patients using brain imaging and then correlated structural brain changes to gene expression. The authors propose topological brain alterations that are associated with TLE and IGE. The topological alterations in TLE and IGE were validated by correlation with the expression of risk genes linked to epilepsies, whereas the expression of risk genes for psychiatric disorders did not correlate to the alterations. While the study is interesting and could lead to a better understanding of pathogenesis in epilepsy, better and more robust evidence should be provided for the association of topological alterations with epilepsy risk factors; the current state of gene expression correlation is rather preliminary.

Major points

1. Correlations of gene expression of epilepsy risk genes and topology predicted by brain imaging are low, with $r \sim 0.2$, and the number of genes used for these correlations is too low. In addition, it is not clear why and how the selection of gene sets for correlation analysis was done. Thus, several measures should be implemented to improve gene expression correlation data before the manuscript could be accepted:

a. To prove the robustness of these correlations, the authors should study alternative datasets of genes that are linked to epilepsy. This is the essential bit that must be added to the story and must clearly validate the topology.

b. The ILAE GWAS study subdivided samples into several categories (please see Suppl. Table 1 in their paper). It seems that Lariviere et al. chose only "Focal Epilepsy, documented hippocampal sclerosis (HS)" to correlate for TLE, since they report:

"We found significant associations between the spatial patterns of multivariate topological alterations in TLE and epilepsy risk gene co-expression levels of hippocampal sclerosis"

Why only these samples were chosen? To be sure to correlate TLE samples from ILAE with TLE samples in the current study? If this is the logic (and this should be then explained in the manuscript), it is clearly underpowered. First of all, the authors should explain "we extracted the most likely genes associated with significant genome-wide loci" – how "most likely" was set? Most importantly, the number of genes equals 3 for "focal epilepsy with HS". This is too low to make robust conclusions. Overall, the following correlations should be done with ILAE dataset:

- The whole ILAE dataset with their TLE and IGE (it seems this is provided as the first panel in figure 3a, however given poor figure legend and text description of what was done it is not clear for the reader)

- All focal ILAE with their TLE

- All focal non-TLE from ILAE with their TLE

- For focal TLE from ILAE with their TLE see comment below*

- All generalized from ILAE with their IGE

*The power of the analysis for psychiatric datasets is clearly higher, e.g. MDD 194 genes and SZ 150 genes. Thus, the authors might consider alternative ways to get more genes for correlation analysis.

One path to explore is to drop the threshold for selection for ILAE GWAS genes and to select more genes, another path is outlined in point 1a – use of alternative datasets, the third path is aggregating epilepsy linked genes across the datasets and measurements (e.g. GWAS, eQTL, omics, etc) for a given type of epilepsy, and there might be other (and better) approaches.

c. For left and right hemispheres, similarly, the only significant correlation, RTLE with focal epilepsy with HS, is underpowered – calculations are based on 3 genes only. Thus, a similar approach as in point 1b should be implemented for left and right topology correlation with gene expression.

d. The authors should provide the exact p values (not only “ $p < 0.05$ ”), across all measures, to evaluate the significance of their data.

e. The description of figure 3 in the text is too short and should be expanded (see also comment below about Nature Communications style), same for figure 3 legend.

2. The format of the manuscript does not fit Nature Communications style. It is very short, with only 3 main figures and many supplementary. While such format might be required for other journals, Nature Communications allows for a longer format with extensive explanation of the rationale, data, conclusions, etc, which helps in understanding a manuscript. Thus, the current manuscript, in particular results and methods sections, should be expanded with a more detailed description of the experiments and the data. For methods, an extended description of computational work should be provided.

3. Reference dataset for gene expression is somewhat outdated (microarray atlas that was published by Allen Brain Institute in 2012). It is a great resource and served the community very well. However, there are comprehensive bulk RNA-seq datasets available, and furthermore multi-regional single cell RNA-seq datasets start to be available. In fact, the implementation of more modern RNA-seq datasets should help in the resolution of their analysis and might improve the robustness of the correlation between topology and gene expression.

Minor points:

1. Although potential application of the identified imaging-genetic signatures in diagnostics was proposed already in the abstract, there is very little discussion about how these signatures could be used in diagnostics.

2. Description of the graphs in figure 1c and the text in the manuscript describing this analysis could be improved. There are significant values above the curves, what exactly do they label, and how do they correspond to the X axis? In addition, is there an explanation for “regional” specificity, e.g. smaller densities for global network alterations in TLE and larger densities for path length in IGE?

Reviewer #3 (Remarks to the Author):

This manuscript describes a network-based analysis of neuroanatomical covariance from multi-center MRI data in epilepsy cohorts through ENIGMA. The authors define global and regional changes in network topology for temporal lobe and idiopathic generalized epilepsies, then compare regional topology changes with postmortem gene expression from the Allen dataset test for spatial correlations with epilepsy-GWAS implicated genes.

The work presented as several strengths: use of a large neuroimaging sample, nice data visualizations, inclusion of some sensible sensitivity analyses. The imaging-transcriptomic questions are also valuable in their testing for dissociable associations across the epilepsy subgroups and psychiatric disorders.

I thought that the following issues would benefit from further consideration. Taken together, these

issues limit the interpretability and therefore the impact of this work. A general theme across these specific issues is a tendency to make several foundational a priori features selections or methodological decisions with little justification or methodological explanation. The main results of this study then depend on the products of these selections/decisions.

1. Justification and contextualization of structural network analyses/results.

- It would be important to understand how the regional variation in graph metric alterations relates to regional variation in the effects of epilepsy on mean CT/volume. In particular, does the CT covariance analysis reveal imaging-transcriptomic associations with risk genes that would be missed by looking at the far simpler metric of regional anatomy change?

- The authors state that "Multivariate comparisons, combining clustering coefficient and path length in TLE relative to controls". More theoretical justification, empirical grounds and methodological details are needed for this step. Why choose these two specific graph metrics to combine? Why combine them in the specific manner used here (and what is this method - congruence of effect size +/- thresholding? If this, then why this vs. alternatives?)? Is it sound to describe the resulting two modes of graph metric change as "regular" and "random"?

2. Transcriptomic associations

- Tests for these would benefit from inclusion of an additional gene set null which addresses different needs to the spin test approach employed. This is discussed in <https://www.biorxiv.org/content/10.1101/2021.02.22.432228v2.abstract>

- Why were common variants from GWAS used rather than, or as well as rare variants? It would be good for the authors to share their reasoning here.

- In their methods, the authors state in relationship to their spatial permutation test that "This framework generates null models of overlap between cortical maps by projecting the spatial coordinates of cortical and subcortical data onto the surface spheres (i.e., the parameter spaces)". This seems to suggest that subcortical data were somehow included in a surface-based spin test with cortical metrics. This is - as far as I know - unprecedented (I may be wrong though). It would be good for the authors to unpack this a bit more - did imaging-transcriptomic associations use both cortical and subcortical information? That alone poses some major questions given the large differences in mean expression of many genes between these two brain compartments. This is a problem for interpretation of the observed correlation, but also - in other ways - for any spin test that ignores the distinction between cortical and subcortical information.

RESPONSE TO THE REVIEWERS (NCOMMS-21-41209)

We would like to thank the Editors for the opportunity to submit a revised manuscript, as well as the Reviewers for their thoughtful evaluations of our paper. We are grateful for the Reviewers' constructive comments and found the suggestions very helpful to improve the quality of our paper. We addressed all suggestions of the Reviewers in a point-by-point fashion and highlighted the corresponding changes in the manuscript in yellow. All revised figures are provided at the end of the present document.

REVIEWER #1

The manuscript by Larivière and colleagues uses ENIGMA consortium data to investigate structural covariance in epilepsy subtypes, as well as the association of altered structural covariance with patterns of brain gene expression for genes that have previously been implicated in risk for epilepsy. The authors report regionally specific structural covariance alterations that differ between IGE and TLE, and they report that this regional specificity co-varies with patterns of expression of genes associated with TLE and IGE respectively. The proposed article has many strengths. The sample size is large (n=578 with temporal lobe epilepsy [TLE], 288 with idiopathic generalized epilepsy [IGE], 1,328 healthy controls) that is well suited for the questions being asked. The use of the Allen Brain Atlas as a transcriptomic reference is not without weaknesses, but it is consistent with current standards in the field. The methods are generally sound in our opinion, in particular the use of covBat is a nice application of a recent methodological development; the treatment of spatial autocorrelation is rigorous; and the analysis suggesting specificity to the epilepsy gene set as opposed to genes associated with other neuropsychiatric disorders is a nice control. The findings are a valuable contribution to the growing field of imaging-genetics. There are however some methodological and interpretative issues that limit our enthusiasm for the paper in its present form. We hope the following comments/suggestions are helpful.

We are grateful to Ayan Mandal and Dr. Aaron Alexander-Bloch for appreciating the strengths of our study and for providing constructive feedback.

(1) The effects of SNPs on gene expression are not always straightforward, and don't always simply affect transcription of the closest gene. The authors should describe how SNPs were mapped to genes for the gene sets they utilize and discuss the limitations of these approaches in the discussion section.

As suggested, we now detailed how the SNPs were mapped to genes in the *Methods* (P. 20):

“Briefly, samples from the ILAE Consortium cohort and population-based datasets were genotyped on SNP arrays and quality-controlled^{105, 106}. Functional mapping and annotation (FUMA) of GWAS¹⁰⁷ was used to map genome-wide significant loci of all epilepsy phenotypes to genes in and around these loci, resulting in a total of 146 mapped genes, with some genes being associated with multiple phenotypes. Our main analysis examined associations between MRI-derived network topological changes in TLE and IGE to gene expression levels of focal epilepsy with hippocampal sclerosis ($n_{\text{genes}}=5$) and generalized epilepsy ($n_{\text{genes}}=43$). To assess specificity of our imaging-transcriptomic associations with focal epilepsy with hippocampal sclerosis and generalized epilepsy phenotypes, we also performed spatial correlations with every other epilepsy phenotype (all epilepsy: $n_{\text{genes}}=16$, focal epilepsy: $n_{\text{genes}}=9$, juvenile myoclonic epilepsy: $n_{\text{genes}}=13$, childhood absence epilepsy: $n_{\text{genes}}=4$).”

And outline limitations of mapping SNPs to gene expression in the *Discussion* (PP. 16-17):

“Limitations of imaging-transcriptomic associations with respect to (i) GWAS-identified genes, (ii) microarray vs. RNA-Seq transcriptomic datasets, and (iii) the mapping of single nucleotide polymorphism (SNP) genotyping on gene expression are hereby highlighted. Firstly, risk genes used in the current study were obtained from a previously published GWAS from the ILAE Consortium that aggregated data from SNP microarrays from 15,212 patients with epilepsy and 29,677 controls³³. The ability of GWAS to identify relevant genes generally scales with overall sample size, and forthcoming studies with larger samples and broader inclusion criteria are expected to expand the catalogue of genes implicated in epilepsy. Rare variants (e.g., causal variants with one rare allele), for instance, are unlikely being tagged by current GWAS-type approaches⁸⁴. Secondly, we derived gene expression from

bulk microarray data obtained from six postmortem donor brains from the AHBA, with predominant cortical and subcortical sampling performed in the left hemisphere. By current standards, the AHBA represents a unique and comprehensive resource to link gene expression and neuroimaging data, offering excellent spatial coverage of nearly the entire human brain and direct mapping of tissue samples to stereotaxic space⁵³. On the other hand, while microarray technology remains a popular and cost-effective approach for transcript profiling, it is limited to interrogating only those genes for which probes are designed⁸⁵. Novel transcriptomics techniques such as RNA-Seq, do not depend on a priori probe selection, and may be more sensitive in identifying genes with low expression and more accurate in detecting expression of common genes⁸⁵. RNA-Seq technology, however, also poses algorithmic and logistical challenges, including the restricted number of samples processed in a single run, elevated costs, data storage requirements, and the absence of an analytical gold standard⁸⁶. Finally, relating GWAS findings to gene expression is a complex process; indeed, GWAS-identified SNPs may occur in non-protein-coding regions⁸⁷, may not always affect transcription of the closest gene, and may implicate genes that are located up to 2 Mbps away⁸⁸. SNPs may also influence several steps of gene expression, particularly messenger RNA (mRNA) splicing, stability, and translation, with their precise functional impact on gene function not being fully understood^{89, 90}.”

(2) Structural covariance relates to several other variables, such as transcriptomic similarity, distance, and structural connectivity. The current understanding and ambiguity regarding biological interpretations of structural covariance should be discussed in either the introduction or discussion. Relatedly, the authors should discuss the question of whether structural covariance network differences primarily reflect neurodevelopment deviations that cause epilepsy, or could they be the consequence of recurrent seizures (or even medications) associated with the condition?

We thank the Reviewers for this comment. We briefly discussed the current understanding of structural covariance in the *Introduction* (P. 4):

“Structural covariance has been associated with several aspects of brain organization and development in both health and disease¹³. In healthy individuals, several studies have shown moderate correspondence with both structural and functional connectivity measures, suggesting partial overlap yet also complementarity of different network mapping techniques^{14, 15, 16}. By comparing cross-sectional covariance networks to longitudinal changes in neurotypical adolescents, prior work has demonstrated a close association of covariance with maturational networks, suggesting that these networks may reflect coordinated trophic processes across the brain^{17, 18, 19}. Furthermore, several studies have pointed to a close association between covariance network layout, heritability, and gene expression, suggesting that genetic factors are also likely reflected in covariance network organization^{20, 21}.”

And further expanded on the biological interpretations of structural covariance changes observed in the common epilepsies in the *Discussion* (P. 15):

“TLE is a complex disorder that is associated with both atypical early neurodevelopment as well as deviations from typical brain aging processes. Growing evidence supports atypical brain development as a potential etiological factor for TLE, with neuroimaging data revealing quantitative changes in cortical folding, hippocampal malrotations, cortical interface blurring, and connectivity alterations in temporo-limbic networks, which may reflect consequences of malformative processes during prenatal stages^{68, 69}. Several histological findings also point to atypical temporo-limbic network formation and maturation^{64, 70}. Following the lifespan, several studies have also suggested interactions between TLE and brain aging, with cross-sectional and longitudinal mesiotemporal volumetry and cortical thickness analysis showing appearance of accelerated brain aging in patients relative to controls^{3, 71, 72, 73}. While often attributed to secondary effects of seizures, these observations may reflect a complex combination of seizure burden, medication load, and psychosocial challenges that medically-intractable patients often face. Notably, when split into short vs. long duration groups, patients with longer duration also exhibited topological regularization primarily in temporo-parietal cortices. This is in line with a previous longitudinal structural covariance analysis in a single centre TLE cohort, which suggests that network alterations may intensify over time²⁴.”

(3) It is known that preponderance of different cell types is a main driver of inter-regional differences in post-mortem gene expression. Is this reflected in differences in the gene sets for TLE and IGE? In other words could the spatial differences in gene expression reflect cell type differences in the gene set?

We thank the Reviewers for this suggestion, and evaluated whether our gene lists were balanced across different cell types using previously published cell type specific gene expression results from Zhu et al., 2018, *Science* (doi.org/10.1126/science.aat8077). This is described in a new analysis on PP. 9-10:

“To ensure that variations in the density of different cell types did not drive transcriptional differences⁵², we evaluated whether our epilepsy-specific gene sets were balanced in terms of their cell-type specificity. We separately calculated average cell-type specificity for 29 transcriptomically distinct cell types for hippocampal sclerosis and generalized epilepsy, based on cell-type specificity estimates derived from 17,093 single-nuclei RNA sequencing (snRNA-seq) samples from the dorsolateral prefrontal cortex of three adult human brains^{53, 54}. In both cases, preponderance of cell types was assessed against null distributions with identical number of genes (see MATERIALS AND METHODS) and showed no significant differences in their cell-type specificity (hippocampal sclerosis: $p_{null} > 0.19$, generalized epilepsy: $p_{null} > 0.11$). Moreover, average cell-type specificity in hippocampal sclerosis and generalized epilepsy gene sets were overall more similar to each other than to null models (all cell-types $p_{null} > 0.065$), with the exception of Endo ($p < 0.05$) and ExN1 ($p < 0.05$).”

With detailed methods on PP. 20-21:

“To evaluate whether interregional differences in postmortem gene expression were driven by varying densities of different cell types, we obtained previously published cell type gene expression results sampled in the adult human dorsolateral prefrontal cortex⁵⁴. Adopting this approach, specificity was assessed in 29 single-nucleus subtypes: Astro1, Astro2, Astro3, Astro4, Endo, ExN1, ExN2a, ExN2b, ExN3e, ExN4, ExN5b, ExN6a, ExN6b, ExN8, InN1a, InN1b, InN1c, InN3, InN4a, InN4b, InN6a, InN6b, InN7, InN8, Microglia, Oligo, OPC1, OPC2, and VSMC. For each TLE- and IGE-related gene set (i.e., hippocampal sclerosis and generalized epilepsy risk genes), we calculated the averaged specificity for each of the 29 cell types. To statistically assess whether different cell types were overexpressed in each epilepsy gene list, we compared their average specificity to null distributions. For each cell type, a null distribution was generated by randomly selecting genes (10,000 iterations) and averaging cell type specificity scores. For consistency, the number of random genes selected was identical to the number of genes in the epilepsy-specific gene set. The empirical (i.e., original) specificity score was compared against the null distribution determined by the ensemble of randomly calculated specificity scores. Using this approach, we also directly compared expression of cell types in hippocampal sclerosis and generalized epilepsy gene sets by comparing the difference in their average specificities (i.e., hippocampal sclerosis – generalized epilepsy) to null distributions [composed of (i) hippocampal sclerosis – randomly selected genes and (ii) generalized epilepsy – randomly selected genes]. Empirical and randomly generated gene lists whose averaged scores equaled zero were discarded from the analysis.”

(4) Given evidence that network edge weight can be associated with topological differences in some network phenotypes, it would be helpful to know if there were differences in total weight between groups.

We thank the Reviewers for this comment. We compared overall mean strength of positive correlations of patients (TLE, IGE) to controls across sites and observed no significant group differences (P. 7):

“Site-specific correlation matrices in TLE, IGE, and healthy controls exhibited similar patterns, with generally strong correlations between bilaterally homologous regions and strong correlations between regions within the same lobe. Overall mean strength of positive correlations across all density thresholds did not differ between individuals with TLE and controls (all $t < 1.86$, $p_{FDR} < 0.13$), but was increased in individuals with IGE relative to controls (all $t < 2.41$, $p_{FDR} < 0.05$).”

Associated methods are described in a new paragraph titled “Strength of cortical thickness and subcortical volume correlations” (P. 19):

“We compared the mean strength of positive correlations (computed from the site-specific thresholded structural covariance networks) in patients relative to controls using Student’s t-tests”

(5) Presumably this information is in the ENIGMA toolbox, but the implementation of the spin test requires a few more details to understand what was done. What was the treatment of subcortical structures? How was the regional parcellation dealt with in the rotated null model?

We apologize for the lack of clarity. We provided additional details about the inclusion of subcortical structures and the use of a regional parcellation in our spin permutation tests on P. 21:

“Here we used a previously published framework for parcellated data¹⁰⁸ and adapted it to include subcortical structures. Briefly, we generated a surface model of all 14 subcortical structures and merged it with the initial reconstructed FreeSurfer cortical surface model. The resulting surface mesh thus comprised adequately placed cortical and subcortical vertices, with original volumetric distances being preserved. We inflated this new combined cortical/subcortical surface model and mapped its vertices to a sphere. In doing so, we obtained the spherical coordinates of each of our 68 cortical regions and 14 subcortical structures¹⁰⁹. We then applied randomly sampled rotations (10,000 repetitions unless specified otherwise) about three axes (x: left-right, y: rostral-caudal, z: dorsal-ventral) at three randomly generated angles, θ_x , θ_y , and $\theta_z \in [0, 2\pi)$ ¹⁰⁸. Following sphere rotation, coordinates of the rotated regions were matched to coordinates of the original regions using Euclidean distance. This matching yielded a mapping from the set of regions to itself, thus allowing the assignment of original values to rotated regions.”

We further cross-validated our spin permutation approach with the variogram-matching technique presented in Burt et al., 2020, *NeuroImage* (<https://doi.org/10.1016/j.neuroimage.2020.117038>), as described in the *Methods* (P. 21) and in a new **FIG. S13**:

“Given that spin permutation models have been predominantly constrained to cortical brain maps, we cross-validated our combined cortical/subcortical spin permutation model to a previously published variogram-matching model^{110, 111}. This method generates surrogate brain maps with matched spatial autocorrelation to that of a target brain, and has been applied to subcortical structures. As recommended¹¹⁰, surrogate maps were generated using surface-based geodesic distance between cortical regions and three-dimensional Euclidean distance between subcortical and cortical/subcortical regions. Variogram-matching null distributions were then generated from randomly shuffling surrogate maps while preserving the distance-dependent correlation between elements of the brain map. As illustrated in FIG. S13, null distributions generated from the spin method were in close agreement with the variogram-matching model and provided nearly identical p-values for spatial correlations, thus supporting the validity of our combined cortical/subcortical spin permutation method.”

(6) How do areas of topological differences relate to any areas of case control differences in mean thickness or volume? If they overlapped this would potentially affect the interpretation. See for example a recent study in schizophrenia discussing this issue <https://ajp.psychiatryonline.org/doi/10.1176/appi.ajp.2019.18040380>. This comparison could be descriptive with a quantitative analysis left for future work.

To address the Reviewers' question, we carried out an additional analysis to examine whether topological changes inferred from structural covariance analysis could be explained by the spatial distributions of atrophied areas in TLE and IGE. We found only minimal overlap between syndrome-related multivariate topological changes and atrophy patterns, suggesting that the inter-regional covariance approach addresses network organization above and beyond regional morphological abnormalities in TLE and IGE (P. 11):

“Compared with univariate mapping of cortical thickness and subcortical volume changes, structural covariance specifically addresses inter-regional structural network organization in TLE and IGE. To test whether TLE-related alterations in covariance patterns are explainable by regional atrophy alone⁵⁶, we first compared atrophy profiles in patients relative to controls using surface-based linear models³. Patterns of atrophy in TLE and IGE were then spatially compared to multivariate (combined clustering and path length) covariance network changes and statistically assessed via non-parametric spin tests³⁷. As in previous studies^{3, 57}, patients with TLE showed profound atrophy in bilateral superior parietal (left/right $p_{FDR}=2.86 \times 10^{-29}/4.50 \times 10^{-27}$), precuneus (left/right $p_{FDR}=3.54 \times 10^{-29}/3.32 \times 10^{-22}$),

precentral (left/right $p_{FDR}=3.95\times 10^{-21}/3.12\times 10^{-20}$), and paracentral (left/right $p_{FDR}=1.75\times 10^{-19}/5.41\times 10^{-18}$) cortices, as well as ipsilateral hippocampus ($p_{FDR}=2.32\times 10^{-186}$) and thalamus ($p_{FDR}=1.25\times 10^{-67}$; FIG. S2A). In contrast, patients with IGE showed predominant atrophy in bilateral precentral cortices (left/right $p_{FDR}=2.94\times 10^{-14}/7.75\times 10^{-12}$) and thalamus (left/right $p_{FDR}=8.63\times 10^{-14}/1.70\times 10^{-14}$; FIG. S2B). The spatial pattern of multivariate topological changes, however, did not closely correspond to areas of atrophy in TLE ($r=0.097$, $p_{spin}=0.21$) nor IGE ($r=-0.061$, $p_{spin}=0.23$), suggesting that covariance changes may not be fully explainable by the spatial distributions of cortical thickness and subcortical volume changes in the same condition.”

Moreover, imaging-genetic associations were significantly weaker when derived from atrophy patterns rather than multivariate topological changes (P. 11):

“Moreover, imaging-transcriptomics associations were significantly weaker when derived from regional atrophy patterns (as opposed to multivariate topological changes) in TLE (correlation with gene expression levels of hippocampal sclerosis: $r=0.041$, $p_{spin/rand}=0.50/0.91$; FIG. S2C) and in IGE (correlation with gene expression levels of generalized epilepsy: $r=-0.083$, $p_{spin/rand}=0.25/0.83$; FIG. S2D).”

Findings are also reported in a new FIG. S2.

(7) Ideally another (and more uniform) parcellation would also be tested. Doing this would require a lot of work and we do not think it's necessary to actually perform this additional analysis, but the reliance on the Desikan parcellation should be discussed as a weakness in the discussion.

We agree with the Reviewers that evaluating our results with another parcellation would be ideal, but reliance on Desikan-Killany atlas remains an inherent limitation of ENIGMA datasets. We discussed this issue on P. 18:

“As data sharing practices can at times be challenging, in part due to privacy and regulatory protection, ENIGMA represents a practical alternative for standardized data processing and anonymized derivative data^{12, 95, 96, 97, 98, 99}. Notably, the Desikan-Killany atlas is widely adopted across ENIGMA Working Groups, thus allowing for comparison of results across initiatives. On the other hand, this parcellation is limited by its relatively coarse granularity (68 cortical regions) and variable parcel sizes. In future studies, replication of our findings with higher-resolution cortical and subcortical parcellations that offer better uniformity in areal definition may help to increase generalizability and specificity.”

(8) Are the disease effects (correspondence between GE and group neuroimaging differences) robust to the exact number of genes nominated as significant for each disease? Since the number of significant GWAS hits is proportional to sample size of studies, this number would be expected to change substantially in the future for all of the disorders considered. It would be interesting to vary the threshold for SNP significance (and thus ultimately the number of genes in the gene set) and see if main results are robust to varying this threshold. This is particularly relevant to the FTE set that only includes 3 genes and might be more susceptible to a false positive result.

We thank the Reviewers for pointing this out. To increase the sample size of epilepsy-related genes, we lowered our criteria for SNP significance and instead included all mapped genes across all phenotypes. This increased our gene sample from $n=19$ to $n=67$, with some genes being associated with multiple epilepsy phenotypes (P. 20).

“Our main analysis examined associations between MRI-derived network topological changes in TLE and IGE to gene expression levels of focal epilepsy with hippocampal sclerosis ($n_{genes}=5$) and generalized epilepsy ($n_{genes}=43$). To assess specificity of our imaging-transcriptomic associations with focal epilepsy with hippocampal sclerosis and generalized epilepsy phenotypes, we also performed spatial correlations with every other epilepsy phenotype (all epilepsy: $n_{genes}=16$, focal epilepsy: $n_{genes}=9$, juvenile myoclonic epilepsy: $n_{genes}=13$, childhood absence epilepsy: $n_{genes}=4$). Due to the low number of genes in some epilepsy phenotypes, our gene lists included all genes mapped in and around significant genome-wide loci (i.e., regions encompassing all SNPs with $p<5\times 10^{-4}$ that were in linkage disequilibrium= $R^2>0.2$), that is, no biological prioritization or pre-selection of genes was performed.”

Moreover, to test whether our imaging-transcriptomic associations were robust to the number of genes in each set, we carried out an additional analysis using non-parametric permutation tests. Briefly, imaging-transcriptomic associations were compared against null models generated by correlating network alterations to transcriptomic maps derived from randomly selected gene sets with identical length. This new analysis, and associated findings, are described in the *Results* (P. 9):

“Additional “random-gene” permutation tests (termed p_{rand}) were performed to (i) test for gene specificity³⁷ and (ii) ensure that imaging-transcriptomic associations were not driven by differences in the number of genes in each syndrome- or disease-specific set (see MATERIALS AND METHODS and FIG. 3A).”

Additional details are provided in the *Methods* (PP. 21-22):

“Random-gene permutation models. Due to differences in the length of the various gene sets (range $n_{genes}=5-152$), we also assessed our imaging-transcriptomic associations against “random-gene” null distributions³⁸. Null distributions were generated by correlating multivariate topological alterations in patients to transcriptomic maps derived from randomly selecting gene sets of equal length from the pool of all $n=12,668$ genes (10,000 iterations). This approach thus allowed us to examine whether our empirical correlation coefficients are significantly larger than those derived from null distributions of identically sized, and randomized, gene sets.”

REVIEWER #2

The manuscript by Lariviere et al. studied changes in brain structure in epileptic patients using brain imaging and then correlated structural brain changes to gene expression. The authors propose topological brain alterations that are associated with TLE and IGE. The topological alterations in TLE and IGE were validated by correlation with the expression of risk genes linked to epilepsies, whereas the expression of risk genes for psychiatric disorders did not correlate to the alterations. While the study is interesting and could lead to a better understanding of pathogenesis in epilepsy, better and more robust evidence should be provided for the association of topological alterations with epilepsy risk factors; the current state of gene expression correlation is rather preliminary.

We thank the Reviewer for the positive evaluation of our work and for the constructive comments to further strengthen our approach.

Major points:

(1) Correlations of gene expression of epilepsy risk genes and topology predicted by brain imaging are low, with $r \sim 0.2$, and the number of genes used for these correlations is too low. In addition, it is not clear why and how the selection of gene sets for correlation analysis was done. Thus, several measures should be implemented to improve gene expression correlation data before the manuscript could be accepted:

We apologize for the confusion. Spatial correlations are performed by comparing two brain maps, for example, multivariate topological changes and mean gene expression. Correlations are then statistically assessed using non-parametric spin permutation tests, which control for spatial autocorrelation of the different maps (Alexander-Bloch et al., 2018, *NeuroImage*; <http://doi.org/10.1016/j.neuroimage.2018.05.070>). To enhance clarity, we provided additional details on our spin permutation tests on P. 21:

“Here we used a previously published framework for parcellated data¹⁰⁸ and adapted it to include subcortical structures. Briefly, we generated a surface model of all 14 subcortical structures and merged it with the initial reconstructed FreeSurfer cortical surface model. The resulting surface mesh thus comprised adequately placed cortical and subcortical vertices, with original volumetric distances being preserved. We inflated this new combined cortical/subcortical surface model and mapped its vertices to a sphere. In doing so, we obtained the spherical coordinates of each of our 68 cortical regions and 14 subcortical structures¹⁰⁹. We then applied randomly sampled rotations (10,000 repetitions unless specified otherwise) about three axes (x: left-right, y: rostral-caudal, z: dorsal-

ventral) at three randomly generated angles, θ_x , θ_y , and $\theta_z \in [0, 2\pi)$ ¹⁰⁸. Following sphere rotation, coordinates of the rotated regions were matched to coordinates of the original regions using Euclidean distance. This matching yielded a mapping from the set of regions to itself, thus allowing the assignment of original values to rotated regions.”

To ensure that the number of genes did not impact the significance of our imaging-transcriptomic associations, we carried out an additional statistical analysis using non-parametric permutation tests. Briefly, imaging-transcriptomic associations were compared against null models generated by correlating network alterations to transcriptomic maps derived from randomly selected gene sets with identical length. This new analysis, and associated findings, are described in the *Results* (P. 9):

“Additional “random-gene” permutation tests (termed p_{rand}) were performed to (i) test for gene specificity³⁸ and (ii) ensure that imaging-transcriptomic associations were not driven by differences in the number of genes in each syndrome- or disease-specific set (see *MATERIALS AND METHODS* and *FIG. 3A*).”

Additional details are provided in the *Methods* (P. 22):

“Random-gene permutation models. Due to differences in the length of the various gene sets (range $n_{genes}=5-152$), we also assessed our imaging-transcriptomic associations against “random-gene” null distributions³⁸. Null distributions were generated by correlating multivariate topological alterations in patients to transcriptomic maps derived from randomly selecting gene sets of equal length from the pool of all $n=12,668$ genes (10,000 iterations). This approach thus allowed us to examine whether our empirical correlation coefficients are significantly larger than those derived from null distributions of identically sized, and randomized, gene sets.”

To enhance clarity, we provided a schema of our non-parametric (spatial) permutation tests in the revised **FIG. 3A**. We further cross-validated our spin permutation approach with the variogram-matching approach presented in Burt et al., 2020, *NeuroImage* (<https://doi.org/10.1016/j.neuroimage.2020.117038>). This method generates surrogate brain maps with matched spatial autocorrelation to that of a target brain, and has been applied to subcortical structures. As depicted in a new **FIG. S11**, null distributions generated from both spin tests and variograms were nearly identical and provided similar p -values for spatial correlations, supporting the validity of our combined cortical/subcortical spin permutation method.

(1a) To prove the robustness of these correlations, the authors should study alternative datasets of genes that are linked to epilepsy. This is the essential bit that must be added to the story and must clearly validate the topology.

As recommended, we assessed specificity of imaging-transcriptomic associations against two additional epilepsy-related datasets: (i) a set of 69 monogenic epilepsy genes (Allen et al., 2017, *Lancet Neurology*; <http://www.genedx.com>) and (ii) 44 genes that are targets of currently used anti-epileptic drugs (Santos et al., 2017, *Nature Reviews Drug Discovery*). Our new findings are described in the *Results* on P. 9:

“To further assess specificity to hippocampal sclerosis (in TLE) and generalized epilepsy (in IGE) genes, we also cross-referenced our network findings with transcriptomic maps derived from (i) genes associated to four additional epilepsy phenotypes, namely: all epilepsy, focal epilepsy, juvenile myoclonic epilepsy, and childhood absence epilepsy³³, (ii) a set of monogenic epilepsy genes from the Epi4K Consortium⁴⁵ and the GeneDX comprehensive epilepsy panel (<http://www.genedx.com>), (iii) genes that are targets of currently used anti-epileptic medications⁴⁶ [...] Network alterations in TLE did not correlate to any other epilepsy subtype (range $r=-0.15-0.13$, all $p_{spin/null}>0.11/0.16$; *FIG. 4*). In contrast, IGE showed additional significant associations with transcriptomic maps derived from all epilepsy ($r=0.37$, $p_{spin/null}=0.0019/0.0032$) and focal epilepsy ($r=0.27$, $p_{spin/null}=0.015/0.034$).”

Findings are also summarized in a new **FIG. 4**.

(1b) The ILAE GWAS study subdivided samples into several categories (please see Suppl. Table 1 in their paper). It seems that Lariviere et al. chose only “Focal Epilepsy, documented hippocampal sclerosis (HS)” to correlate for TLE, since they report: “We found significant associations between the spatial patterns of multivariate topological alterations in TLE and epilepsy risk gene co-expression levels of

hippocampal sclerosis". Why only these samples were chosen? To be sure to correlate TLE samples from ILAE with TLE samples in the current study? If this is the logic (and this should be then explained in the manuscript), it is clearly underpowered. First of all, the authors should explain "we extracted the most likely genes associated with significant genome-wide loci" – how "most likely" was set? Most importantly, the number of genes equals 3 for "focal epilepsy with HS". This is too low to make robust conclusions.

Overall, the following correlations should be done with ILAE dataset:

- The whole ILAE dataset with their TLE and IGE (it seems this is provided as the first panel in figure 3a, however given poor figure legend and text description of what was done it is not clear for the reader)
- All focal ILAE with their TLE
- All focal non-TLE from ILAE with their TLE
- For focal TLE from ILAE with their TLE see comment below*
- All generalized from ILAE with their IGE

*The power of the analysis for psychiatric datasets is clearly higher, e.g. MDD 194 genes and SZ 150 genes. Thus, the authors might consider alternative ways to get more genes for correlation analysis. One path to explore is to drop the threshold for selection for ILAE GWAS genes and to select more genes, another path is outlined in point (1a) – use of alternative datasets, the third path is aggregating epilepsy linked genes across the datasets and measurements (e.g. GWAS, eQTL, omics, etc) for a given type of epilepsy, and there might be other (and better) approaches.

As suggested by the Reviewer, we performed an exhaustive correlation analysis within the ILAE dataset, assessing spatial correlations between network alterations in TLE/IGE and transcriptomic maps from all six epilepsy subtypes (see our response to comment 1a). Associations with the two principal epilepsy subtypes (i.e., focal epilepsy with hippocampal sclerosis and generalized epilepsy) are now summarized in the revised **FIG. 3B** (see figure in our response to comment 1). Spatial associations with the other four epilepsy subtypes are presented in a new **FIG. 4** (see our response to comment 1a).

In addition to our added statistical analysis approach which assesses robustness of correlations against randomly generated gene sets of equal length (see comment 1), we also increased our gene sample by revising our inclusion criteria for the ILAE gene sets. Specifically, we lowered our criteria for SNP significance to include all mapped genes across all phenotypes, regardless of their biological prioritization scores. This increased our gene sample from $n=19$ to $n=67$, with some genes being associated with multiple epilepsy phenotypes (P. 20).

"Our main analysis examined associations between MRI-derived network topological changes in TLE and IGE to gene expression levels of focal epilepsy with hippocampal sclerosis ($n_{genes}=5$) and generalized epilepsy ($n_{genes}=43$). To assess specificity of our imaging-transcriptomic associations with focal epilepsy with hippocampal sclerosis and generalized epilepsy phenotypes, we also performed spatial correlations with every other epilepsy phenotype (all epilepsy: $n_{genes}=16$, focal epilepsy: $n_{genes}=9$, juvenile myoclonic epilepsy: $n_{genes}=13$, childhood absence epilepsy: $n_{genes}=4$). Due to the low number of genes in some epilepsy phenotypes, our gene lists included all genes mapped in and around significant genome-wide loci (i.e., regions encompassing all SNPs with $p < 5 \times 10^{-4}$ that were in linkage disequilibrium= $R^2 > 0.2$), that is, no biological prioritization or pre-selection of genes was performed."

We reiterate that all spatial correlations are performed by comparing two brain maps (with the same number of parcels), and, thus, power is homogenized across analyses. Differences in the number of genes per sets, however, may affect GWAS sensitivities. We addressed this issue in our response to comment 1, but nevertheless acknowledge this as a potential issue in the *Discussion* (P. 17):

"Notably, these distinctive imaging-transcriptomic associations were robust and remained significant after comparison against null distributions derived from randomly selected gene sets of equal length. Nonetheless, differences in the number of genes in each gene lists may have contributed to variability in gene expression profiles."

(1c) For left and right hemispheres, similarly, the only significant correlation, RTLE with focal epilepsy with HS, is underpowered – calculations are based on 3 genes only. Thus, a similar approach as in point (1b) should be implemented for left and right topology correlation with gene expression.

We applied the same approaches that are described in our responses to comments **1**, **1a**, and **1b** for left and right TLE analyses. These new findings are described in the *Results* on PP. 11-12:

“Differences in multivariate topological changes between left and right TLE marginally affected their associations with epilepsy- (FIG. S5) and disease-related (FIG. S6) risk genes; spatial correlation with expression levels of genes previously associated to hippocampal sclerosis was only significant in left, but not right TLE. Left TLE also showed a significant association to the ‘all epilepsy’ subtype ($r=0.25$, $p_{spin/rand}=0.022/0.032$). Network alterations in right TLE, on the other hand, correlated with transcriptomic maps derived from generalized epilepsy genes ($r=0.17$, $p_{spin}=0.048$) and bipolar disorder ($r=0.20$, $p_{spin}=0.018$); these correlations, however, did not survive comparison against randomly selected genes, $p_{rand}>0.14$.”

Results are also summarized in revised **FIGS. S3-6**.

(1d) The authors should provide the exact p values (not only “ $p<0.05$ ”), across all measures, to evaluate the significance of their data.

As requested by the Reviewer, we provided exact *p*-values and the corresponding statistics for all results throughout the manuscript.

(1e) The description of figure 3 in the text is too short and should be expanded (see also comment below about Nature Communications style), same for figure 3 legend.

As requested, we expanded the in-text descriptions (both in the *Results* and *Methods*) of imaging-transcriptomic analyses and figures (now **FIGS. 3, 4, and 5**) as well as associated figure legends.

(2) The format of the manuscript does not fit Nature Communications style. It is very short, with only 3 main figures and many supplementary. While such format might be required for other journals, Nature Communications allows for a longer format with extensive explanation of the rationale, data, conclusions, etc, which helps in understanding a manuscript. Thus, the current manuscript, in particular results and methods sections, should be expanded with a more detailed description of the experiments and the data. For methods, an extended description of computational work should be provided.

We thank the Reviewer for this excellent suggestion and expanded descriptions in every section within the revised manuscript (overall >50% increase). Moreover, we included two new main Figures, five new Supplementary Figures, and expanded our figure legends for additional clarity.

(3) Reference dataset for gene expression is somewhat outdated (microarray atlas that was published by Allen Brain Institute in 2012). It is a great resource and served the community very well. However, there are comprehensive bulk RNA-seq datasets available, and furthermore multi-regional single cell RNA-seq datasets start to be available. In fact, the implementation of more modern RNA-seq datasets should help in the resolution of their analysis and might improve the robustness of the correlation between topology and gene expression.

In the revised manuscript, we acknowledged limitations related to microarray datasets, and recognized the potential of RNA-seq analysis for the identification of new genetic variants in epilepsy. We nevertheless also maintained that the AHBA offers several advantages over more modern RNA-seq datasets, including high brain coverage and stereotaxic mapping of tissue samples (PP. 16-17):

“Limitations of imaging-transcriptomic associations with respect to (i) GWAS-identified genes, (ii) microarray vs. RNA-Seq transcriptomic datasets, and (iii) the mapping of single nucleotide polymorphism (SNP) genotyping on gene expression are hereby highlighted. Firstly, risk genes used in the current study were obtained from a previously published GWAS from the ILAE Consortium that

aggregated data from SNP microarrays from 15,212 patients with epilepsy and 29,677 controls³³. The ability of GWAS to identify relevant genes generally scales with overall sample size, and forthcoming studies with larger samples and broader inclusion criteria are expected to expand the catalogue of genes implicated in epilepsy. Rare variants (e.g., causal variants with one rare allele), for instance, are unlikely being tagged by current GWAS-type approaches⁸⁴. Secondly, we derived gene expression from bulk microarray data obtained from six postmortem donor brains from the AHBA, with predominant cortical and subcortical sampling performed in the left hemisphere. By current standards, the AHBA represents a unique and comprehensive resource to link gene expression and neuroimaging data, offering excellent spatial coverage of nearly the entire human brain and direct mapping of tissue samples to stereotaxic space⁵³. On the other hand, while microarray technology remains a popular and cost-effective approach for transcript profiling, it is limited to interrogating only those genes for which probes are designed⁸⁵. Novel transcriptomics techniques such as RNA-Seq, do not depend on a priori probe selection, and may be more sensitive in identifying genes with low expression and more accurate in detecting expression of common genes⁸⁵. RNA-Seq technology, however, also poses algorithmic and logistical challenges, including the restricted number of samples processed in a single run, elevated costs, data storage requirements, and the absence of an analytical gold standard⁸⁶. Finally, relating GWAS findings to gene expression is a complex process; indeed, GWAS-identified SNPs may occur in non-protein-coding regions⁸⁷, may not always affect transcription of the closest gene, and may implicate genes that are located up to 2 Mbps away⁸⁸. SNPs may also influence several steps of gene expression, particularly messenger RNA (mRNA) splicing, stability, and translation, with their precise functional impact on gene function not being fully understood^{89,90}. Replication of our findings in more comprehensive, RNA-Seq gene expression datasets may hold significant promise for stratification and effective treatment that can be targeted to the individual patients based on their genetic profile. Once the barriers to widespread use of RNA-Seq are overcome, our understanding of the genetic architecture of the epilepsies will significantly evolve, with the reported risk genes likely being expanded and refined as more genomic and transcriptomic data become available.”

Minor points:

(1) Although potential application of the identified imaging-genetic signatures in diagnostics was proposed already in the abstract, there is very little discussion about how these signatures could be used in diagnostics.

We thank the Reviewer for pointing this out. We now briefly discussed the potential of imaging-transcriptomic associations in guiding epilepsy diagnosis on P. 16.

“From a clinical standpoint, these findings represent a glance of the different pathophysiological anomalies in temporal lobe and generalized epilepsies, and may lead to possible imaging-transcriptomic applications for improved patient stratification. For instance, the genetically-linked network maps identified herein may increase diagnostic sensitivity in both TLE and IGE, while also pointing to different, syndrome-specific, genetically-mediated etiologies. Alternatively, as these subnetwork alterations were associated with syndrome-related risk genes, our findings could provide a foundation for future research aiming to explore whether targeted assessments of these subnetworks can help to discriminate gene variant carriers vs. non-carriers, thus potentially enhancing diagnostics and treatment calibration.”

(2) Description of the graphs in figure 1c and the text in the manuscript describing this analysis could be improved. There are significant values above the curves, what exactly do they label, and how do they correspond to the X axis? In addition, is there an explanation for “regional” specificity, e.g. smaller densities for global network alterations in TLE and larger densities for path length in IGE?

We included an inset in **FIG. 1C** to describe the meaning of the labels and revised the figure caption as follows:

“Student’s t-tests were performed at each density value, comparing global measures in patients (TLE or IGE) to controls; bold asterisks indicate $p_{FDR} < 0.1$, semi-transparent asterisks indicate $p_{uncorr} < 0.05$.”

The observed trends for global changes in higher network density thresholds could indicate that network alterations in IGE target weaker connections (which would be otherwise removed at smaller densities). This is now mentioned in the *Results* (P. 7):

“In contrast, IGE patients showed, on average, similar overall clustering coefficient relative to controls, but marginal decreases in overall path length at higher network densities ($p_{\text{uncorr}} < 0.05$ at $K=0.31-0.50$; FIG. 1C), possibly targeting weaker interregional correlations.”

REVIEWER #3

This manuscript describes a network-based analysis of neuroanatomical covariance from multi-center MRI data in epilepsy cohorts through ENIGMA. The authors define global and regional changes in network topology for temporal lobe and idiopathic generalized epilepsies, then compare regional topology changes with postmortem gene expression from the Allen dataset test for spatial correlations with epilepsy-GWAS implicated genes. The work presented as several strengths: use of a large neuroimaging sample, nice data visualizations, inclusion of some sensible sensitivity analyses. The imaging-transcriptomic questions are also valuable in their testing for dissociable associations across the epilepsy subgroups and psychiatric disorders. I thought that the following issues would benefit from further consideration. Taken together, these issues limit the interpretability and therefore the impact of this work. A general theme across these specific issues is a tendency to make several foundational a priori features selections or methodological decisions with little justification or methodological explanation. The main results of this study then depend on the products of these selections/decisions.

We are grateful to the Reviewer for appreciating the strengths and significance of our work, and for providing constructive feedback.

(1) Justification and contextualization of structural network analyses/results.

(1a) It would be important to understand how the regional variation in graph metric alterations relates to regional variation in the effects of epilepsy on mean CT/volume. In particular, does the CT covariance analysis reveal imaging-transcriptomic associations with risk genes that would be missed by looking at the far simpler metric of regional anatomy change?

To address the Reviewer’s questions, we carried out an additional analysis to examine whether topological changes inferred from structural covariance analysis could be explained by the spatial distributions of atrophied areas in TLE and IGE. We showed minimal overlap between syndrome-specific multivariate topological changes and atrophy patterns, suggesting that our covariance approach specifically addresses the network organization underlying whole-brain morphological abnormalities in TLE and IGE (P. 11):

“Compared with univariate mapping of cortical thickness and subcortical volume changes, structural covariance specifically addresses inter-regional structural network organization in TLE and IGE. To test whether TLE-related alterations in covariance patterns are explainable by regional atrophy alone⁵⁶, we first compared atrophy profiles in patients relative to controls using surface-based linear models³. Patterns of atrophy in TLE and IGE were then spatially compared to multivariate (combined clustering and path length) covariance network changes and statistically assessed via non-parametric spin tests³⁷. As in previous studies^{3,57}, patients with TLE showed profound atrophy in bilateral superior parietal (left/right $p_{\text{FDR}}=2.86 \times 10^{-29}/4.50 \times 10^{-27}$), precuneus (left/right $p_{\text{FDR}}=3.54 \times 10^{-29}/3.32 \times 10^{-22}$), precentral (left/right $p_{\text{FDR}}=3.95 \times 10^{-21}/3.12 \times 10^{-20}$), and paracentral (left/right $p_{\text{FDR}}=1.75 \times 10^{-19}/5.41 \times 10^{-18}$) cortices, as well as ipsilateral hippocampus ($p_{\text{FDR}}=2.32 \times 10^{-186}$) and thalamus ($p_{\text{FDR}}=1.25 \times 10^{-67}$; FIG. S2A). In contrast, patients with IGE showed predominant atrophy in bilateral precentral cortices (left/right $p_{\text{FDR}}=2.94 \times 10^{-14}/7.75 \times 10^{-12}$) and thalamus (left/right $p_{\text{FDR}}=8.63 \times 10^{-14}/1.70 \times 10^{-14}$; FIG. S2B). The spatial pattern of multivariate topological changes, however, did not closely correspond to areas of atrophy in TLE ($r=0.097$, $p_{\text{spin}}=0.21$) nor IGE ($r=-0.061$, $p_{\text{spin}}=0.23$), suggesting that covariance changes may not be fully explainable by the spatial distributions of cortical thickness and subcortical volume changes in the same condition.”

Moreover, imaging-genetic associations were significantly weaker when derived from atrophy patterns rather than multivariate topological changes (P. 11):

“Moreover, imaging-transcriptomics associations were significantly weaker when derived from regional atrophy patterns (as opposed to multivariate topological changes) in TLE (correlation with gene expression levels of hippocampal sclerosis: $r=0.041$, $p_{spin/rand}=0.50/0.91$; FIG. S2C) and in IGE (correlation with gene expression levels of generalized epilepsy: $r=-0.083$, $p_{spin/rand}=0.25/0.83$; FIG. S2D).”

Findings are also reported in a new **FIG. S2**.

(1b) The authors state that “Multivariate comparisons, combining clustering coefficient and path length in TLE relative to controls”. More theoretical justification, empirical grounds and methodological details are needed for this step. Why choose these two specific graph metrics to combine? Why combine them in the specific manner used here (and what is this method - congruence of effect size +/- thresholding? If this, then why this vs. alternatives?)? Is it sound to describe the resulting two modes of graph metric change as "regular" and "random"?

We thank the Reviewer for these questions. We justified the use of multivariate statistical analysis in the *Methods* (P. 20):

“To signify an overall load of anomalies, we subsequently compared the aggregate of clustering coefficient and path length differences in patients relative to controls using multivariate surface-based linear models. This approach allowed topological changes to be described in a compact manner, and consequently, enabled spatial associations with brain maps of gene expression (see section on Transcriptomic associations). Moreover, by statistically combining clustering coefficient and path length, we leveraged their covariance to obtain a substantial gain in sensitivity, and thus, unveiled subthreshold network properties not readily identified in a single graph theoretical metric.”

The ratio of clustering to path length indexes the small-worldness of a network, and thus, allowed us to characterize whether, and how, patients deviate from this optimal organization. For completeness, in the revised version, we also reported group-differences in global and regional small-world indices. Findings are presented in a new **FIG. S1**. We provided additional details to justify our choice of these three graph theoretical metrics of interest on P. 7:

“To characterize the topology of structural covariance networks, we computed three fundamental and widely used graph-theoretical parameters⁴²: (i) mean clustering coefficient, to quantify local network efficiency, (ii) mean path length, to index global efficiency, and (iii) mean small-world index, to quantify the interaction of both local and global efficiency. Notably, the interplay between clustering coefficient and path length can categorize network topology into regular or random, and consequently assess deviations from an optimal small-world architecture. At either extreme, regular, or “lattice-like,” networks have high clustering and path length, whereas random networks have low clustering and path length. On the other hand, small-world networks are neither completely random nor regular, but have high clustering and low path length and, thus, reflect a locally and globally efficient organization (FIG. 1B)⁴³.”

We also further clarified this approach in the *Methods* (P. 19):

“[...] we computed three global metrics using standard formulas^{42, 103}: (i) mean clustering coefficient, which quantifies the tendency for brain regions to be locally interconnected with neighboring regions, (ii) mean path length, which quantifies the mean minimum number of edges (i.e., connection between two regions) that separate any two regions in the network, and (iii) small-world index (mean clustering coefficient divided by mean path length), which quantifies both local and global properties. These two metrics, along with their combination (i.e., small-world index) are the most widely used graph theoretical parameters to describe the topology of complex networks.”

(2) Transcriptomic associations

(2a) Tests for these would benefit from inclusion of an additional gene set null which addresses different needs to the spin test approach employed. This is discussed in <https://www.biorxiv.org/content/10.1101/2021.02.22.432228v2.abstract>

We thank the Reviewer for this suggestion. We employed a similar approach to that described in Wei et al., 2022, *Hum Brain Mapp* to test for gene specificity and robustness of our associations with the number of genes in each subtype/disease. Briefly, imaging-transcriptomic associations were compared against null models generated by correlating network alterations to transcriptomic maps derived from randomly selected gene sets with identical length. This new analysis, and associated findings, are described in the *Results* (P.9):

“Additional “random-gene” permutation tests (termed p_{rand}) were performed to (i) test for gene specificity³⁸ and (ii) ensure that imaging-transcriptomic associations were not driven by differences in the number of genes in each syndrome- or disease-specific set (see MATERIALS AND METHODS and FIG. 3A).”

Additional details are provided in the *Methods* (P. 22):

“Random-gene permutation models. Due to differences in the length of the various gene sets (range $n_{genes}=5-152$), we also assessed our imaging-transcriptomic associations against “random-gene” null distributions³⁸. Null distributions were generated by correlating multivariate topological alterations in patients to transcriptomic maps derived from randomly selecting gene sets of equal length from the pool of all $n=12,668$ genes (10,000 iterations). This approach thus allowed us to examine whether our empirical correlation coefficients are significantly larger than those derived from null distributions of identically sized, and randomized, gene sets.”

To enhance clarity, we provided a schema of our nonparametric (spatial) permutation tests in the revised **FIG. 3A**.

(2b) Why were common variants from GWAS used rather than, or as well as rare variants? It would be good for the authors to share their reasoning here.

In the revised manuscript, we assessed the specificity of our imaging-transcriptomic associations against two additional epilepsy-related gene datasets: (i) a set of 69 monogenic epilepsy genes and (ii) 44 genes that are targets of currently used anti-epileptic drugs. These gene lists contain rare coding variants, including SLC2A1, GABRA1, and GABRG2. Our new findings are described in the *Results*, on P. 9:

“To further assess specificity to hippocampal sclerosis (in TLE) and generalized epilepsy (in IGE) genes, we also cross-referenced our network findings with transcriptomic maps derived from (i) genes associated to four additional epilepsy phenotypes, namely: all epilepsy, focal epilepsy, juvenile myoclonic epilepsy, and childhood absence epilepsy³³, (ii) a set of monogenic epilepsy genes from the Epi4K Consortium⁴⁵ and the GeneDX comprehensive epilepsy panel (<http://www.genedx.com>), (iii) genes that are targets of currently used anti-epileptic medications⁴⁶ [...] Network alterations in TLE did not correlate to any other epilepsy subtype (range $r=-0.15-0.13$, all $p_{spin/null}>0.11/0.16$; FIG. 4). In contrast, IGE showed additional significant associations with transcriptomic maps derived from all epilepsy ($r=0.37$, $p_{spin/null}=0.0019/0.0032$) and focal epilepsy ($r=0.27$, $p_{spin/null}=0.015/0.034$).”

Our main imaging-transcriptomic analysis (**FIG. 3**), however, was restricted to common variants due to inherent limitations of SNP genotyping in tagging rare variants. We now also discussed this as a limitation, along with the potential of RNA sequencing studies to overcome this problem, on P. 17:

“The ability of GWAS to identify relevant genes generally scales with overall sample size, and forthcoming studies with larger samples and broader inclusion criteria are expected to expand the catalogue of genes implicated in epilepsy. Rare variants (e.g., causal variants with one rare allele), for instance, are unlikely being tagged by current GWAS-type approaches⁸⁴. [...] Novel transcriptomics techniques such as RNA-Seq, do not depend on a priori probe selection, and may be more sensitive in identifying genes with low expression and more accurate in detecting expression of common genes⁸⁵. [...] Once the barriers to widespread use of RNA-Seq are overcome, our understanding of the genetic

architecture of the epilepsies will significantly evolve, with the reported risk genes likely being expanded and refined as more genomic and transcriptomic data become available.”

(2c) In their methods, the authors state in relationship to their spatial permutation test that “This framework generates null models of overlap between cortical maps by projecting the spatial coordinates of cortical and subcortical data onto the surface spheres (i.e., the parameter spaces)”. This seems to suggest that subcortical data were somehow included in a surface-based spin test with cortical metrics. This is - as far as I know - unprecedented (I may be wrong though). It would be good for the authors to unpack this a bit more - did imaging-transcriptomic associations use both cortical and subcortical information? That alone poses some major questions given the large differences in mean expression of many genes between these two brain compartments. This is a problem for interpretation of the observed correlation, but also - in other ways - for any spin test that ignores the distinction between cortical and subcortical information.

We apologize for the lack of clarity regarding the spin permutation test procedure. We provided additional details about the inclusion of subcortical structures and the use of a regional parcellation in our spin permutation tests on P. 21:

“Here we used a previously published framework for parcellated data ¹⁰⁸ and adapted it to include subcortical structures. Briefly, we generated a surface model of all 14 subcortical structures and merged it with the initial reconstructed FreeSurfer cortical surface model. The resulting surface mesh thus comprised adequately placed cortical and subcortical vertices, with original volumetric distances being preserved. We inflated this new combined cortical/subcortical surface model and mapped its vertices to a sphere. In doing so, we obtained the spherical coordinates of each of our 68 cortical regions and 14 subcortical structures ¹⁰⁹. We then applied randomly sampled rotations (10,000 repetitions unless specified otherwise) about three axes (x: left-right, y: rostral-caudal, z: dorsal-ventral) at three randomly generated angles, θ_x , θ_y , and $\theta_z \in [0, 2\pi)$ ¹⁰⁸. Following sphere rotation, coordinates of the rotated regions were matched to coordinates of the original regions using Euclidean distance. This matching yielded a mapping from the set of regions to itself, thus allowing the assignment of original values to rotated regions.”

We further cross-validated our spin permutation approach with the variogram-matching approach presented in Burt et al., 2020, *NeuroImage* (<https://doi.org/10.1016/j.neuroimage.2020.117038>), as described in the *Methods* (P. 21) and in a new **FIG. S13**:

“Given that spin permutation models have been predominantly constrained to cortical brain maps, we cross-validated our combined cortical/subcortical spin permutation model to a previously published variogram-matching model ^{110, 111}. This method generates surrogate brain maps with matched spatial autocorrelation to that of a target brain, and has been applied to subcortical structures. As recommended ¹¹⁰, surrogate maps were generated using surface-based geodesic distance between cortical regions and three-dimensional Euclidean distance between subcortical and cortical/subcortical regions. Variogram-matching null distributions were then generated from randomly shuffling surrogate maps while preserving the distance-dependent correlation between elements of the brain map. As illustrated in FIG. S13, null distributions generated from the spin method were in close agreement with the variogram-matching model and provided nearly identical p-values for spatial correlations, thus supporting the validity of our combined cortical/subcortical spin permutation method.”

Cortical and subcortical gene expression data were combined to allow for spatial imaging-transcriptomic associations (our main structural covariance analysis was performed on combined cortical and subcortical imaging data). To account for differences in microarray expression between these two compartments, we performed separate normalization procedures for each structural designation, as recommended by Arnatkevičiūtė et al., 2019, *NeuroImage*. This is now clarified in the *Methods* (P. 20):

“To account for known differences in microarray expression between broad structural compartments (e.g., cortex vs. subcortex) ¹⁰⁴, normalization procedures were performed separately for cortical and subcortical structures ⁵³.”

REVISED FIGURES

a | Construction of group- and site-specific structural covariance networks

b | Graph theoretical parameters and topological properties

c | Global network alterations in the common epilepsies

Figure 1. Structural covariance networks in the common epilepsies. (a) Schematic showing the construction of group- and site-specific structural covariance networks from morphometric correlations. (b) Two graph theoretical parameters characterized network topology: clustering coefficient, which measures connection density among neighboring nodes (*orange*) and path length, which measures the number of shortest steps between any two given nodes (*purple*). The interplay between clustering coefficient and path length can describe three distinct topological organizations: regular networks with high clustering and path length (*left*), small-world networks with high clustering and low path length (*middle*), and random networks with low clustering and path length (*right*). (c) Global differences in clustering coefficient (*left*) and path length (*right*) between TLE and HC (*top*) and between IGE and HC (*bottom*) are plotted as a function of network density. Increased small-worldness (*i.e.*, increased clustering and decreased path length) was observed in individuals with TLE, whereas individuals with IGE showed decreases in clustering and path length, suggesting a more random configuration. Student's *t*-tests were performed at each density value, comparing global measures in patients (TLE or IGE) to controls; bold asterisks indicate $p_{FDR} < 0.1$, semi-transparent asterisks indicate $p_{uncorr} < 0.05$. Thin lines represent data from individual sites.

a | Statistical testing of imaging-transcriptomics associations

b | Transcriptomic associations with TLE and IGE epilepsy risk genes

Figure 3. Imaging-transcriptomic associations. (a) Approaches for statistical testing of imaging-transcriptomic associations. Gene expression data for a subset of phenotype- or disease-specific genes are averaged and spatially compared to the patterns of multivariate topological changes in TLE and IGE independently. Spatial correlations are statistically assessed using (i) spatial permutation models, which preserve the spatial autocorrelation of brain maps (10,000 permutations), and (ii) permutation models, which generate null distributions from randomised gene expression data with identical length as the original gene set (10,000 permutations). (b) Gene expression levels associated with two distinct epilepsy subtypes (focal epilepsy with hippocampal sclerosis and generalized epilepsy) were mapped to cortical and subcortical surface templates and spatially compared to patterns of multivariate topological alterations (which combined clustering and path length; see FIG. 2). In TLE, spatial associations between microarray data and multivariate topological changes were strongest for expression levels of hippocampal sclerosis genes ($r=0.33$, $p_{spin}=0.0028$). On the other hand, in IGE, spatial associations were strongest for expression levels of generalized epilepsy genes ($r=0.31$, $p_{spin}=0.0032$). Both TLE- and IGE-specific imaging-transcriptomic associations were robust against null distributions of effects based on selecting random genes from the full gene set (TLE: $p_{rand}=0.0030$, IGE: $p_{rand}=0.018$).

Figure 4. Relations between epilepsy gene expression and network topology. Gene expression levels associated with (i) all other epilepsy subtypes (all epilepsy, focal epilepsy, juvenile myoclonic epilepsy, and childhood absence epilepsy), (ii) monogenic epilepsy, and (iii) anti-epileptic drug targets were mapped to cortical and subcortical surface templates. Spatial correlations were performed between each of these transcriptomic maps and the patterns of multivariate topological alterations in TLE and IGE. In IGE, spatial associations between microarray data and multivariate topological changes were significant for expression levels of all epilepsy genes ($r=0.37$, $p_{spin}=0.0019$) and focal epilepsy ($r=0.27$, $p_{spin}=0.015$). In TLE, network associations did not correlate with any other epilepsy-related transcriptomic maps.

Figure 5. Relations between disease-related gene expression and network topology. Gene expression levels associated with six common neuropsychiatric conditions and/or comorbidities of epilepsy (attention deficit/hyperactivity disorder, autism spectrum disorder, bipolar disorder, major depressive disorder, migraine, and schizophrenia) were mapped to cortical and subcortical surface templates. Spatial correlations were performed between each of these transcriptomic maps and the patterns of multivariate topological alterations in TLE and IGE. In IGE, a spatial association between microarray data and multivariate topological changes was significant for expression levels of major depression disorder genes ($r=0.19$, $p_{spin}=0.015$). This association, however, did not survive correction against a null distribution of effects based on selecting random genes ($p_{rand}=0.18$). In TLE, network associations did not correlate with any other disease-related transcriptomic maps.

a | Global and regional small-world alterations in TLE

b | Global and regional small-world alterations in IGE

Figure S1. Small-world organization in TLE and IGE. (a) Global (top) and regional (bottom) differences in small-world index between temporal lobe epilepsy (TLE) and healthy controls (HC) are plotted as a function of network density. Student's *t*-tests comparing TLE patients to controls were performed at each density value; trends for increased small-worldness was observed in individuals with TLE ($p_{\text{uncorr}} < 0.05$). Bold asterisks indicate $p_{\text{FDR}} < 0.1$, semi-transparent asterisks indicate $p_{\text{uncorr}} < 0.05$. Thin lines represent data from individual sites. Cohen's *d* effect sizes pointed to overall increases in regional small-world index in TLE. (b) Global (top) and regional (bottom) differences in small-world index between idiopathic generalized epilepsy (IGE) and HC are plotted as a function of network density. Student's *t*-tests comparing IGE patients to controls were performed at each density value; overall small-world organization was virtually identical in individuals with IGE and HC. Thin lines represent data from individual sites. Cohen's *d* effect sizes pointed to slight increases and decreases in regional small-world index in IGE.

Figure S2. Associations between cortical/subcortical atrophy and epilepsy risk genes. (a) Cortical thickness and subcortical volume reductions in TLE, compared to healthy controls, spanned bilateral superior parietal (left/right $p_{FDR}=2.86\times 10^{-29}/4.50\times 10^{-27}$), precuneus (left/right $p_{FDR}=3.54\times 10^{-29}/3.32\times 10^{-22}$), precentral (left/right $p_{FDR}=3.95\times 10^{-21}/3.12\times 10^{-20}$), and paracentral (left/right $p_{FDR}=1.75\times 10^{-19}/5.41\times 10^{-18}$) cortices, as well as ipsilateral hippocampus ($p_{FDR}=2.32\times 10^{-186}$) and thalamus ($p_{FDR}=1.25\times 10^{-67}$). (b) In contrast, grey matter cortical and subcortical atrophy in IGE, relative to controls, was more subtle and affected predominantly bilateral precentral cortices (left/right $p_{FDR}=2.94\times 10^{-14}/7.75\times 10^{-12}$) and thalamus (left/right $p_{FDR}=8.63\times 10^{-14}/1.70\times 10^{-14}$). Negative \log_{10} -transformed FDR-corrected p -values are shown. (c) Imaging-transcriptomics associations were significantly weaker when derived from regional atrophy patterns (as opposed to multivariate topological changes) in TLE (correlation with gene expression levels of hippocampal sclerosis: $r=0.041$, $p_{spin/rand}=0.50/0.91$). (d) Similarly, imaging-transcriptomics associations were also significantly weaker when derived from regional atrophy patterns in IGE (correlation with gene expression levels of generalized epilepsy: $r=-0.083$, $p_{spin/rand}=0.25/0.83$).

a | Global and regional network alterations in left TLE

b | Global and regional network alterations in right TLE

Figure S3. Structural covariance networks in left (LTLE) and right (RTLE) TLE. (a) Global differences in clustering coefficient (*top left*) and path length (*top right*) between left LTLE and healthy controls (HC) are plotted as a function of network density. Increased small-worldness (increased clustering coefficient, decreased path length) was observed in individuals with left TLE. Student's *t*-tests comparing LTLE patients to controls were performed at each density value; bold asterisks indicate $p_{FDR} < 0.05$, semi-transparent asterisks indicate $p_{FDR} < 0.1$. Thin lines represent data from individual sites. Multivariate topological differences in left TLE were primarily observed in bilateral fronto-temporal cortices, and revealed a regular network configuration (increased clustering and path length). (b) Global differences in clustering coefficient (*top left*) and path length (*top right*) between RTLE and HC are plotted as a function of network density. Increased small-worldness (increased clustering coefficient, decreased path length) was observed in individuals with right TLE. Student's *t*-tests comparing RTLE patients to controls were performed at each density value; bold asterisks indicate $p_{FDR} < 0.05$, semi-transparent asterisks indicate $p_{FDR} < 0.1$. Thin lines represent data from individual sites. Multivariate topological changes in right TLE were primarily observed in bilateral fronto-temporal cortices and the hippocampus, and revealed a widespread regular network configuration.

a | Global network alterations in left vs. right TLE

b | Regional network alterations in left vs. right TLE

Figure S4. Structural covariance networks in left (LTLE) vs. right (RTLE) TLE. (a) Global differences in clustering coefficient (*left*) and path length (*right*) between LTLE and RTLE are plotted as a function of network density. No significant difference was observed. Student's *t*-tests were performed at each density value; bold asterisks indicate $p_{FDR} < 0.05$, semi-transparent asterisks indicate $p_{FDR} < 0.1$. Thin lines represent data from individual sites. (b) Trends for multivariate topological changes in LTLE vs. RTLE were observed in ipsilateral middle frontal gyrus and entorhinal cortex as well as contralateral calcarine sulcus. Compared to the other subcohort, LTLE showed network regularization (increased clustering and path length) in sensorimotor cortices, whereas RTLE showed widespread network regularization.

Figure S5. Relations between epilepsy gene expression and network topology. Spatial correlations were performed between gene expression levels associated with every epilepsy subtype (focal epilepsy with hippocampal sclerosis, generalized epilepsy, all epilepsy, focal epilepsy, juvenile myoclonic epilepsy, and childhood absence epilepsy), (ii) monogenic epilepsy, and (ii) anti-epileptic drug targets and the patterns of multivariate topological alterations in left (LTLE) and right (RTLE) TLE. In left TLE, spatial associations between microarray data and multivariate topological changes were significant for expression levels of focal epilepsy with hippocampal sclerosis ($r=0.32$, $p_{\text{spin/rand}}=0.0050/0.0045$) and all epilepsy ($r=0.25$, $p_{\text{spin/rand}}=0.022/0.032$). In right TLE, network associations only correlated with transcriptomic maps of generalized epilepsy ($r=0.32$, $p_{\text{spin}}=0.048$), but did not survive correction against a “random-gene” null distribution ($p_{\text{rand}}=0.19$).

Figure S6. Relations between disease-related gene expression and network topology. Spatial correlations were performed between gene expression levels associated with six common neuropsychiatric conditions and/or comorbidities of epilepsy (attention deficit/hyperactivity disorder, autism spectrum disorder, bipolar disorder, major depressive disorder, migraine, and schizophrenia) and the patterns of multivariate topological alterations in left (LTLE) and right (RTLE) TLE. In left TLE network associations did not correlate with any other epilepsy-related transcriptomic maps. In right TLE, network associations correlated with transcriptomic maps of bipolar disorder ($r=0.20$, $p_{\text{spin}}=0.018$), but did not survive correction against a “random-gene” null distribution ($p_{\text{rand}}=0.14$).

Figure S13. Cross-validation of cortical/subcortical spin permutation test. Statistical significance of the imaging-transcriptomic spatial correlations from **Fig. 3B** was assessed using the spin permutation approach (pink and blue distributions) and variogram-matching model (gray distribution). Both methods generated similar distributions and nearly identical p -values.

REVIEWER COMMENTS

Reviewer #1 (Remarks to the Author):

The authors have addressed all of our concerns in this revision.

Reviewer #2 (Remarks to the Author):

The authors did an impressive work during revision, the manuscript is significantly improved and all my points have been addressed.

Reviewer #3 (Remarks to the Author):

The authors have fully responded to all my comments raised at initial review.

Reviewer #4 (Remarks to the Author):

The authors have adequately addressed my concerns, and I have no further comments.